# A repurposed AMP binding domain reveals mitochondrial protein AMPylation as a regulator of cellular metabolism

Abner Gonzalez [1], Alex Pon[1], Kelly Servage[2,3], Krzysztof Pawłowski [2,3], Yan Han [4] & Anju Sreelatha [1,5] ✉

Protein AMPylation, the covalent addition of adenosine monophosphate (AMP) to protein substrates, has been known as a post translational modification for over 50 years. Research in this field is largely underdeveloped due to the lack of tools that enable the systematic identification of AMPylated substrates. Here, we address this gap by developing an enrichment technique to isolate and study AMPylated proteins using a nucleotide-binding protein, hinT. Cryo-EM reconstruction of an AMPylated protein bound to hinT provides a structural basis for AMP selectivity. Using structure guided mutagenesis, we optimize enrichment to identify novel substrates of the evolutionarily conserved AMPylase, Selenoprotein O. We show that mammalian Selenoprotein O regulates metabolic flux through AMPylation of key mitochondrial proteins including glutamate dehydrogenase and pyruvate dehydrogenase. Our findings highlight the broader significance of AMPylation, an emerging post translational modification with critical roles in signal transduction and disease pathology. Furthermore, we establish a powerful enrichment platform for the discovery of novel AMPylated proteins to study the mechanisms and significance of protein AMPylation in cellular function.

Post-translational modification (PTM) of proteins expands the functional diversity and complexity of the proteome. PTMs such as phosphorylation, acetylation, and methylation are well-conserved from bacteria to humans, highlighting their fundamental role in modifying the physical and chemical properties of proteins to regulate cellular signaling. AMPylation (adenylylation) is an evolutionarily conserved PTM characterized by the covalent addition of adenosine monophosphate (AMP) to the hydroxyl side chains of amino acids in protein substrates.

The first instance of AMPylation was identified in the regulation of bacterial nitrogen metabolism in 1967[1,2]. Glutamine synthetase adenyltransferase, GlnE (GS-ATase), catalyzes AMPylation of glutamine synthetase, GlnA, to inhibit synthesis of glutamine during conditions of nitrogen abundance[1–4]. GlnE harbors the nucleotidyltransferase fold similar to the *Legionella pneumophila* AMPylase, SidM, which catalyzes AMPylation of the host Rab1 GTPases during bacterial infection[5]. The Fic-domain containing proteins from several pathogenic bacteria catalyze AMPylation of the eukaryotic Rho GTPases to manipulate host cell signaling during infection[5–7]. *Vibrio parahaemolyticus* VopS and *Histophilus somni* IbpA catalyze AMPylation of Rho GTPases to disrupt the host cytoskeleton[6,7]. These examples underscore the role of protein AMPylation in bacterial metabolism and pathogenicity.

Until recently, the only eukaryotic enzyme known to catalyze AMPylation was the Fic-domain containing enzyme, FicD, which AMPylates heat shock protein, BiP, in the endoplasmic reticulum (ER)[8–11]. FicD regulates the unfolded protein response through

[1]Department of Physiology, University of Texas Southwestern Medical Center, Dallas, TX, USA. [2]Department of Molecular Biology, University of Texas Southwestern Medical Center, Dallas, TX, USA. [3]Howard Hughes Medical Institute, University of Texas Southwestern Medical Center, Dallas, Texas, USA. [4]Department of Biophysics, University of Texas Southwestern Medical Center, Dallas, TX, USA. [5]Charles and Jane Pak Center for Mineral Metabolism and Clinical Research, University of Texas Southwestern Medical Center, Dallas, TX, USA. ✉e-mail: anju.sreelatha@utsouthwestern.edu

AMPylation of BiP in murine pancreas and liver during physiological stress[12–14]. We discovered a second mammalian enzyme with AMPylation activity, Selenoprotein O (SelO), which is localized in the mitochondria[15]. SelO is conserved in *E. coli* and *S. cerevisiae,* where it catalyzes AMPylation to protect cells from oxidative damage and cell death[15]. Notably, SelO-mediated protein AMPylation promotes melanoma metastasis in an immunocompetent mouse melanoma model by regulating oxidative stress[16]. Thus, protein AMPylation is emerging as a novel signaling mechanism with relevance to diverse pathobiology.

Relative to PTMs such as phosphorylation, the breadth of protein AMPylation is underexplored due to the low abundance of AMPylated proteins and limited tools for investigation in cells. Protein AMPylation can be detected using radiolabeled [α−$^{32}$P] ATP or immunoblotting with AMPylation-specific antibodies[17–20]. Although mass spectrometry (MS) fragmentation patterns can be used to detect AMPylation, analysis requires enrichment of AMPylated proteins, given the low abundance of AMPylation in cells[21]. Due to the limited sensitivity and specificity of immunoprecipitation with the antibodies, chemical-proteomic approaches using ATP analogs are mainly used to enrich for AMPylated proteins from lysates[17]. ATP derivative probes labeled on the N6 position with an alkyne or azide permit copper(I) catalyzed azide-alkyne cycloaddition (CuAAC or click chemistry) for their subsequent ligation with an affinity tag or fluorophore[17,22]. Using N6-propargyl ATP, substrates of bacterial and mammalian Fic AMPylases have been detected in vitro[17,22–24]. We utilized a biotinylated ATP analog to demonstrate that SelO catalyzes AMPylation of multiple proteins involved in metabolism and redox homeostasis in vitro[15,25]. However, the negative charge on the phosphates of ATP limits the uptake and utility of these probes in cells. To overcome this limitation, Kielkowski et al. recently developed a cell-permeable pronucleotide, N6-propargyl adenosine phosphoramidate, that is amenable to subsequent click chemistry[26]. The pronucleotide is metabolized to N6-propargyl ATP in the cytosol, where it must compete with endogenous ATP present in millimolar concentrations. Hence, this probe may be limited in sensitivity and permeability into subcellular organelles. Furthermore, the N6 position that is commonly used for chemical handles may be important for the nucleotide binding and activity of some AMPylases, thereby limiting the application of these ATP analogs. These studies highlight the need for novel strategies to identify AMPylated proteins in cells, both to shed light on the functional importance of AMPylation and to facilitate the discovery of novel AMPylases.

Here, we develop an enrichment strategy based on histidine triad nucleotide binding protein (hinT) for the study of protein AMPylation. We solve the cryo-EM structure of an AMPylated protein bound to the nucleotide binding pocket of hinT to provide structural insights into the mechanism of substrate recognition and specificity. Furthermore, we optimize hinT for the detection and isolation of AMPylated proteins from bacterial and mammalian cells to investigate the functional importance of AMPylation. Our strategy led us to the identification of previously unknown AMPylated mitochondrial proteins and revealed their functional importance in altering metabolic flux. We anticipate that these studies will pave the way for a better understanding of AMPylation in cellular regulation and underscore the critical function of SelO in mitochondrial biology.

## Results

### Identification of an AMP binding domain to enrich for AMPylation

We developed a strategy to enrich for AMPylated proteins using the histidine triad nucleotide-binding protein, hinT. Interestingly, hinT was initially isolated in 1997 from rabbit heart cytosol using adenosine agarose affinity chromatography[27,28]. Although its physiological substrates are still unknown, in vitro characterization of hinT revealed high-affinity purine nucleotide binding and phosphoramidase

activity[29]. Mutation of an active site nucleophilic histidine to asparagine (human H112N; *E. coli* H101N) results in a 150-fold increase in the binding affinity of human hinT1 to AMP from 59 μM to 385 nM[30]. Based on the previously reported binding affinity and specificity of hinT, we reasoned that hinT could be repurposed as an AMP-binding domain to enrich for AMPylated proteins.

To utilize hinT in an enrichment strategy, we tagged *E. coli* hinT H101N (hinT$^{HN}$) with glutathione S-transferase (GST) and immobilized to glutathione agarose beads (Fig. 1A). We then used GST-hinT$^{HN}$ to isolate proteins from *E. coli* and identified a prominent band around 55 kDa. Tandem mass spectrometry (MS/MS) analysis identified this protein as GlnA (Supplementary Table 1). Notably, the 55 kDa protein was absent upon enrichment from lysates of cells deleted for the GlnA AMPylase, GlnE, or GlnA itself (Fig. 1B). Furthermore, MS/MS analysis of enriched GlnA revealed tryptic peptides corresponding to AMPylation of a single amino acid, tyrosine-398, in agreement with published reports[31,32] (Supplementary Fig. 1a).

There are only two known enzymes, GlnE and SelO, that catalyze AMPylation in *E. coli*. Although GlnE-mediated AMPylation of GlnA can be detected from cell lysates without the need for enrichment, SelO-mediated AMPylation is far less abundant. Deletion of SelO did not alter the AMPylation of GlnA observed in cell lysates or upon enrichment (Fig. 1B, C).

*E. coli* SelO harbors a conserved metal-binding DFG motif where D256 binds Mg$^{2+}$[15,33]. Mutation of the metal-binding aspartate results in an inactive SelO enzyme. Although SelO lacks the catalytic HRD motif commonly found in kinases, where the aspartate acts as a catalytic base, SelO contains a highly conserved valine in the position of the aspartate[33]. Mutation of V242 to alanine increases the AMPylation activity of SelO through unknown mechanisms. To assess if GST-hinT$^{HN}$ binds to SelO substrates, we analyzed enrichment of AMPylated proteins from *E. coli* lysates overexpressing wild-type SelO, the inactive mutant SelO D256A, or the hyperactive mutant SelO V242A. GST-hinT$^{HN}$, but not GST alone, enriched for SelO-mediated AMPylated proteins (Fig. 1D, E). MS analysis of the enriched proteins identified several potential substrates of *E. coli* SelO in addition to previously identified substrates (Supplementary Data 1) (Supplementary Fig. 1b and 1c)[15]. Collectively, these studies establish a novel method to enrich for AMPylated substrates from cellular lysates.

### Structural basis for AMPylated protein binding to hinT

*E. coli* and *H. sapiens* homologs of hinT exist as homodimers with a conserved active site motif, HxHxHxx, where x denotes a hydrophobic residue[34,35]. To investigate the structural mechanisms of AMPylated protein binding to hinT$^{HN}$, we used AMPylated GlnA (GlnA-AMP) as a model substrate. GlnA forms a dodecamer composed of two hexameric rings that is ideal for cryo-electron microscopy (cryo-EM) analysis[36].

The cryo-EM structure of hinT H101N bound to GlnA revealed up to six hinT dimers bound to the external surface of GlnA dodecamer (Fig. 2A, Supplementary Figs. 2, 3, and Supplementary Table 2). Focused refinement of hinT$^{HN}$ depicted the flexible loops of GlnA-AMP extending into the binding pocket of hinT$^{HN}$ (Fig. 2B, C). The hinT$^{HN}$ dimer binds two GlnA molecules, with each hinT$^{HN}$ subunit interacting slightly differently (Fig. 2D and Supplementary Fig. 4a). Some interactions are similar between the subunits, such as Ile32 interacting with the adenine group. In addition, Met113 extends into the binding pocket of the neighboring hinT, coming into close proximity with the AMPylated amino acid, Tyr398, of GlnA-AMP. His103 of both subunits are next to the phosphate group of AMP, likely making polar contacts with the oxygens of the phosphate group.

Previous X-ray crystallography studies of *E.coli* hinT H101A with guanosine monophosphate, GMP (PDB 3N1T), demonstrate that GMP

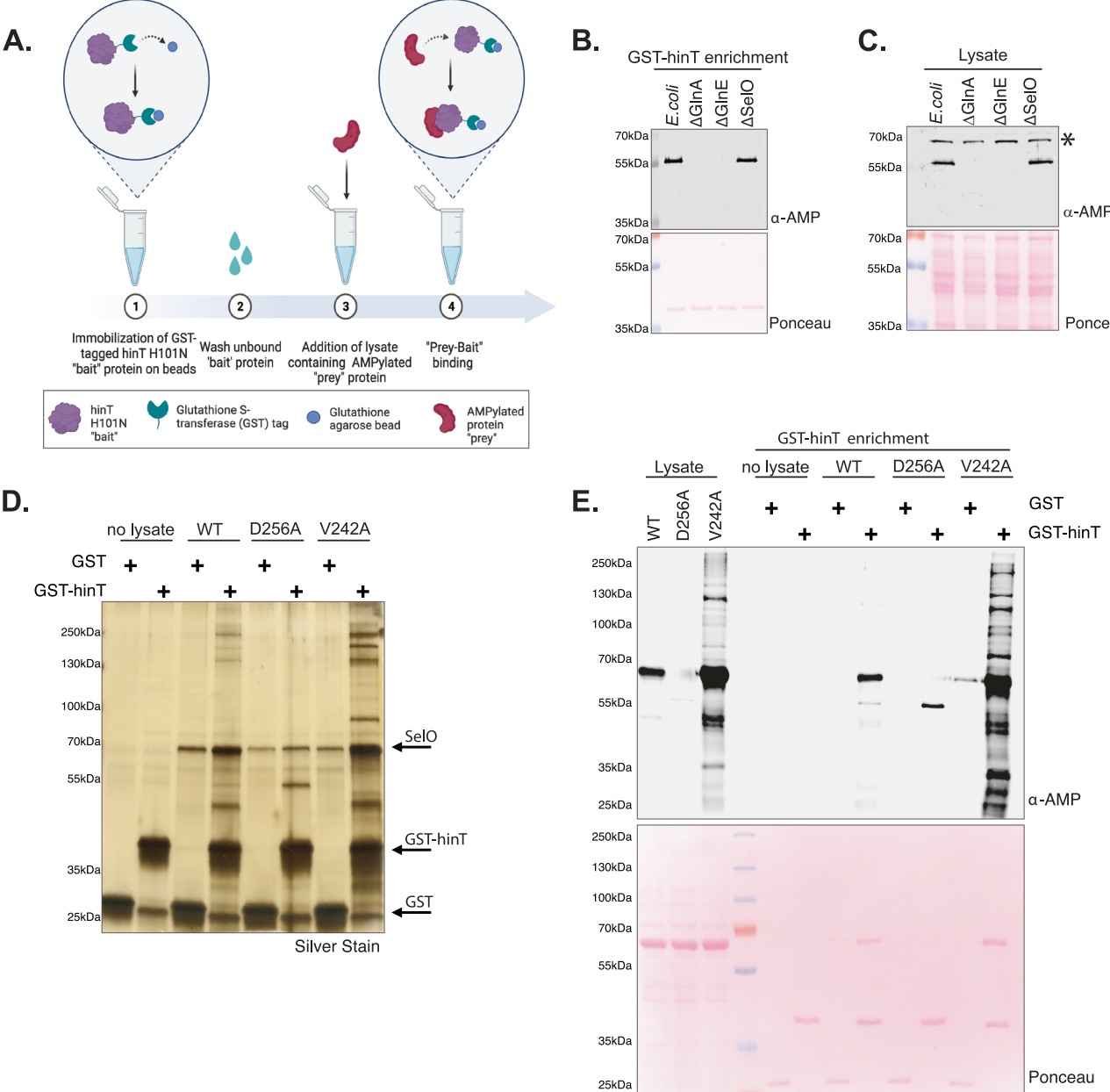

**Fig. 1 | Engineered AMP binding domain enriches AMPylated proteins from *E. coli* lysates. A** Schematic representation of the AMPylated protein enrichment strategy using GST-hinT H101N immobilized on glutathione agarose as bait to pulldown interacting AMPylated proteins in lysates. Created in BioRender. Sreelatha, A. (2025) https://BioRender.com/6bpqez2. **B** Protein immunoblotting of proteins enriched using GST-hinT H101N from wild-type *E. coli*, glnA knockout (ΔGlnA), glnE knockout (ΔGlnE), or SelO knockout (ΔSelO) *E. coli* lysates. The ponceau-stained membrane is shown. **C** Protein immunoblots of cell lysates from wild-type *E.coli*, glnA knockout (ΔGlnA), glnE knockout (ΔGlnE), or SelO knockout

(ΔSelO) *E. coli*. *denotes a non-specific band observed at 70 kDa. The ponceau-stained membrane is shown. **D** SDS-PAGE analysis of proteins enriched using GST or GST-hinT H101N from *E. coli* lysates expressing wild-type (WT) *E.coli* SelO, inactive D256A SelO, or hyperactive V242A SelO. The enriched proteins were visualized by silver stain. **E** Protein immunoblots of proteins enriched using GST or GST-hinT H101N from *E. coli* lysates expressing WT SelO, inactive D256A SelO, or hyperactive V242A SelO. The ponceau-stained membrane is shown. Results depicted in Fig. 1B–E are representative of at least 3 independent experiments.

binds in a similar orientation as the AMPylated tyrosine of GlnA (Supplementary Fig. 4b). The superposition of GMP and AMPylated GlnA bound to hinT[HN] reveal a pocket of hydrophobic residues which coordinates the purine nucleobase, while the conserved His103 and His105 stabilize the phosphate moiety[30,34].Notably, our structural analysis revealed a shift in the ribose and phosphate moieties of AMPylated tyrosine, in comparison to the GMP, which may facilitate the binding of AMPylated proteins relative to free nucleotides (Supplementary Fig. 4b). These studies define the amino acid interactions that affect AMPylated protein binding and build a platform for

structure guided mutagenesis to engineer variants of hinT with enhanced affinity for AMPylated proteins.

## Structure guided mutagenesis and homolog screening reveals hinT mutants with increased affinity

HinT is highly conserved from bacteria to humans, including the hydrophobic and polar residues which form the adenine and phosphate binding pockets[28].In contrast, the C terminus of hinT is highly variable among homologs and contributes to substrate specificity and catalytic function in vitro[34,37]. Furthermore, the

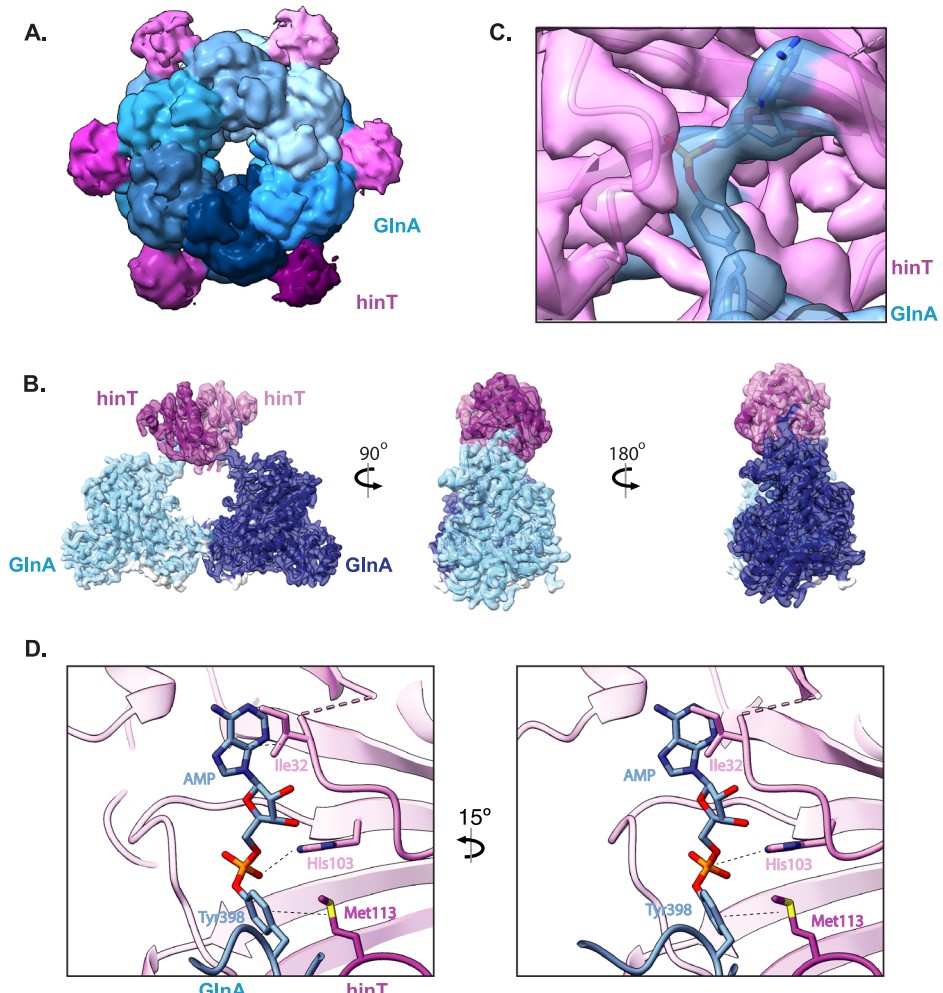

**Fig. 2 | Structure of AMPylated GlnA - hinT H101N complex reveals amino acids at the binding interface. A** Cryo-EM density map representation of the GlnA-hinT H101N complex. The dodecamer GlnA is in blue and the dimeric hinT H101N is in pink. Map obtained from the 3D classification step, for which resolution is not reported by gold-standard FSC. The map is shown at a threshold of 0.023 to better display the density corresponding to hinT. **B** Enlarged image of cryo-EM density map representation of GlnA-hinT binding interface. AMPylated Tyr-398 present in each of the two GlnA molecules extends into the active site of the dimeric hinT H101N. GlnA subunits colored in dark and light blue. hinT dimer is in magenta and light pink. Map resolution 3.5 Å. **C** Zoomed in view of the cryo-EM density map representation of GlnA Tyr398-AMP within the hinT H101N binding pocket. GlnA is colored in blue. hinT is in pink. **D** Enlarged image of the nucleotide binding pocket of hinT H101N, highlighting the hinT amino acid interactions with AMPylated Tyr398 of GlnA.

variability in the C termini of hinT homologs is hypothesized to drive species-specific interactions with substrates[37]. Sequence analysis revealed that the *E. coli* and *H. sapiens* hinT homologs share 49% identity with considerable divergence at the C terminus (Fig. 3A). Interestingly, the C terminus of the hinT homolog from thermophilic bacteria, *Thermaerobacter marianensis*, is more similar to the human homolog than to the *E.coli* homolog, with the latter sharing 44% identity with *T. marianensis*.

To investigate the importance of the C terminus and optimize AMPylated protein enrichment, we determined the binding affinity of *E. coli*, *T. marianensis*, and the human homolog of hinT$^{HN}$ to a model AMPylated protein, sucA-AMP, using biolayer interferometry (BLI) (Fig. 3B–E and Supplementary Fig. 5a–d). We observed that *E. coli* hinT$^{HN}$ binds to sucA-AMP with a $K_D$ of 27 nM (Fig. 3B, E, F). The multiple AMPylation sites present in sucA precluded direct fitting of kinetic parameters; therefore, we restricted our analysis to the steady-state dose response. *T. marianensis* and human homologs of hinT$^{HN}$ displayed a ~ 3-fold increase in binding affinity, with $K_D$ values of 9 nM and 10 nM, respectively (Fig. 3C–F). However, we did not observe binding of unmodified sucA to hinT homologs (Supplementary

Fig. 5d). These data suggest that hinT homologs exhibit varying affinities to AMPylated protein.

Based on published studies examining the role of the C-terminal loop in AMP binding, and the absorbance at 260 nm of recombinant hinT$^{HN}$ proteins, we hypothesized that hinT$^{HN}$ homologs may copurify with AMP nucleotides[37]. HPLC-MS analysis of purified proteins demonstrates that human hinT$^{HN}$ copurified with ~100 fold more AMP in comparison to the *E. coli* and *T. marianensis* homologs of hinT (Supplementary Fig. 5e). Concomitantly, isothermal titration calorimetry (ITC) showed that AMP bound to human hinT$^{HN}$ with a $K_D$ of 279 nM (Supplementary Fig. 6a). *T. marianensis* and *E. coli* homologs of hinT$^{HN}$ displayed a $K_D$ of 3 μM and 69 μM, respectively (Supplementary Fig. 6b,c). Collectively, these studies identify variants of hinT for the optimization of AMPylated protein enrichment. Although human hinT$^{HN}$ displayed high affinity for AMPylated proteins, its strong binding to AMP nucleotides may limit the enrichment of AMPylated proteins due to competition with nucleotides present in cellular lysates. *T. marianensis* and *E. coli* homologs of hinT$^{HN}$ offer balanced functionality due to their relatively low binding affinity for AMP and high affinity for AMPylated proteins.

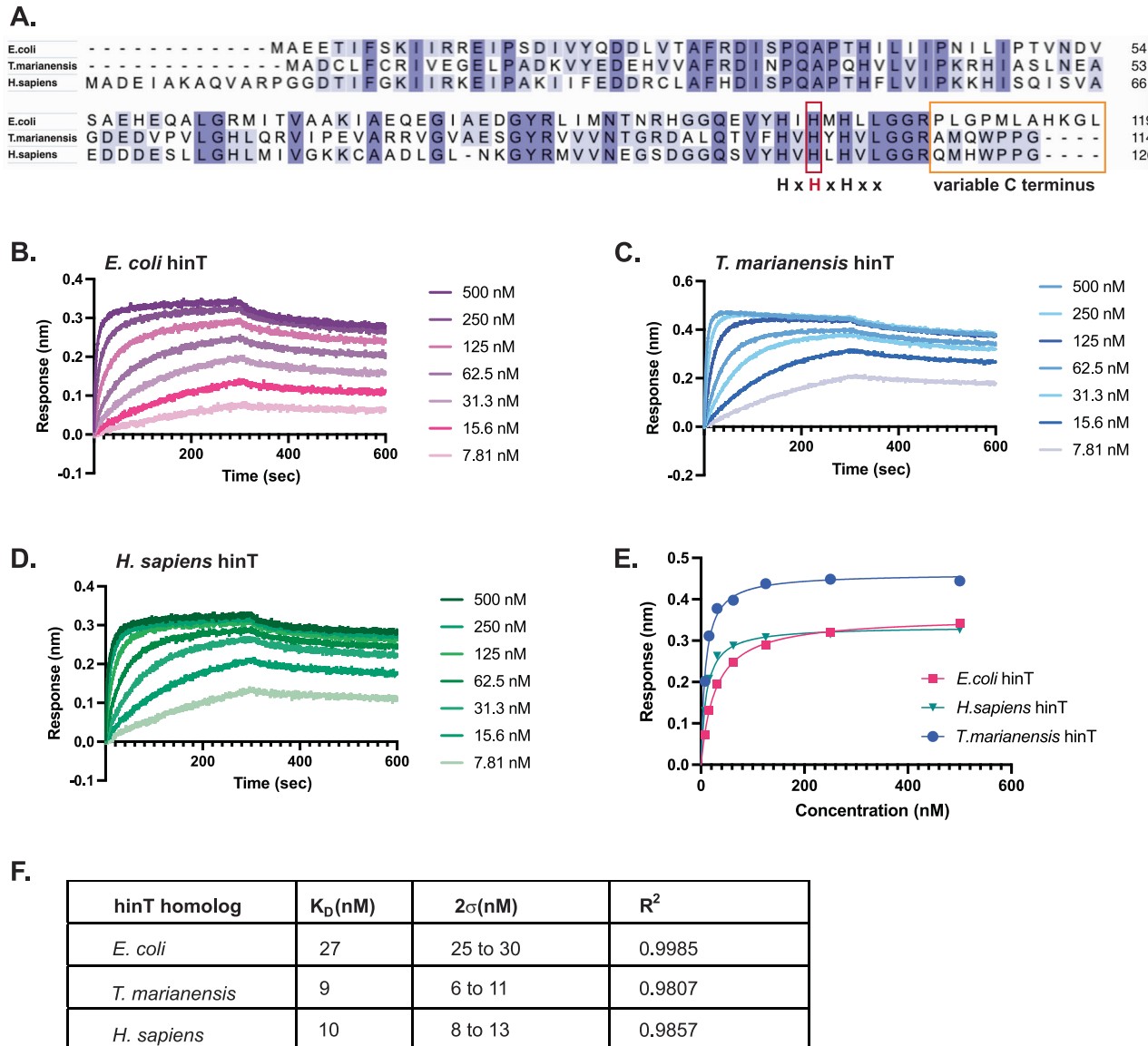

**Fig. 3 | Structure guided mutagenesis and homolog screening reveals hinT mutants with increased affinity. A** Multiple sequence alignment of *E. coli*, *T. marianensis*, and *H. sapiens* hinT. Conserved amino acids are highlighted in light purple (two of three homologs) and dark purple (in all three homologs).The conserved histidine triad motif is noted with the histidine corresponding to *E. coli* His101 labeled in red. Amino acids present in the variable C termini of homologs are boxed in orange. **B**–**D**Representative BLI sensorgrams depicting the binding response of serial dilutions of 500 nM to 7.8 nM AMPylated sucA to immobilized GST-*E. coli* hinT (**B**), GST-*T. marianensis* hinT (**C**), and GST-*H. sapiens* hinT (**D**). **E** Steady state binding response of sucA-AMP to immobilized GST-hinT homologs from BLI experiment. **F** Binding affinities measured from the steady state binding response of GST-hinT homologs to sucA-AMP. The $K_D$ values were determined from steady state binding responses as a non-linear regression curve and reported with a 95% confidence interval. Source data are provided as a Source Data file.

To identify amino acid residues in the binding pocket that may contribute to substrate specificity, we compared the structures of *E. coli* (PDB 3N1S)[34], human (PDB 5KLZ)[38], and *T. marianensis* (AlphaFold)[39] homologs of hinT (Supplementary Fig. 7a–d). Notably, Glu96, which is in close proximity to the phosphate moiety in *E. coli* hinT, is not conserved in human and *T. marianensis* homologs of hinT (Fig. 3A and Supplementary Fig. 7a–d). *T. marianensis* hinT contains a threonine at the position of *E. coli* Glu96, while human hinT has a serine at the same position (Fig. 3A). Both serine and threonine can engage in polar interactions with the phosphate group. However, the negatively charged glutamate may induce repulsion, contributing to the reduced affinity for AMPylated proteins observed with *E. coli* hinT (Supplementary Fig. 7a–d). By identifying specific amino acids or structural changes that enhance binding affinity, the hinT homologs present opportunities for improved performance in the detection and purification of AMPylated proteins.

## hinT variants display varying specificity and sensitivity to AMPylated substrates

To analyze the binding and detection limits of hinT homologs to previously characterized AMPylated proteins, we performed dot blot assays using serial dilutions of AMPylated proteins and their corresponding non-AMPylated controls. Rab1-AMP, sodA-AMP, and GlnA-AMP are AMPylated on tyrosine residues, whereas Rac1-AMP contains an AMPylated threonine[5–7]. Membranes spotted with serial dilutions of proteins were incubated with GST, GST-tagged *E. coli* hinT H101N (GST-Ec hinT^HN), *T. marianensis* hinT H100N (GST-Tm hinT^HN), *H. sapiens* hinT H112N (GST-Hs hinT^HN), or *E. coli* hinT H101N E96S (GST-Ec hinT^HN, E96S). We mutated Glu96 of *E. coli* hinT to mimic the serine found in the human homolog, which has high affinity for AMPylated proteins. Mutation of E96S resulted in a modest increase in binding affinity of *E. coli* hinT^HN to $K_D$ of 21 nM (Supplementary Fig. 8).

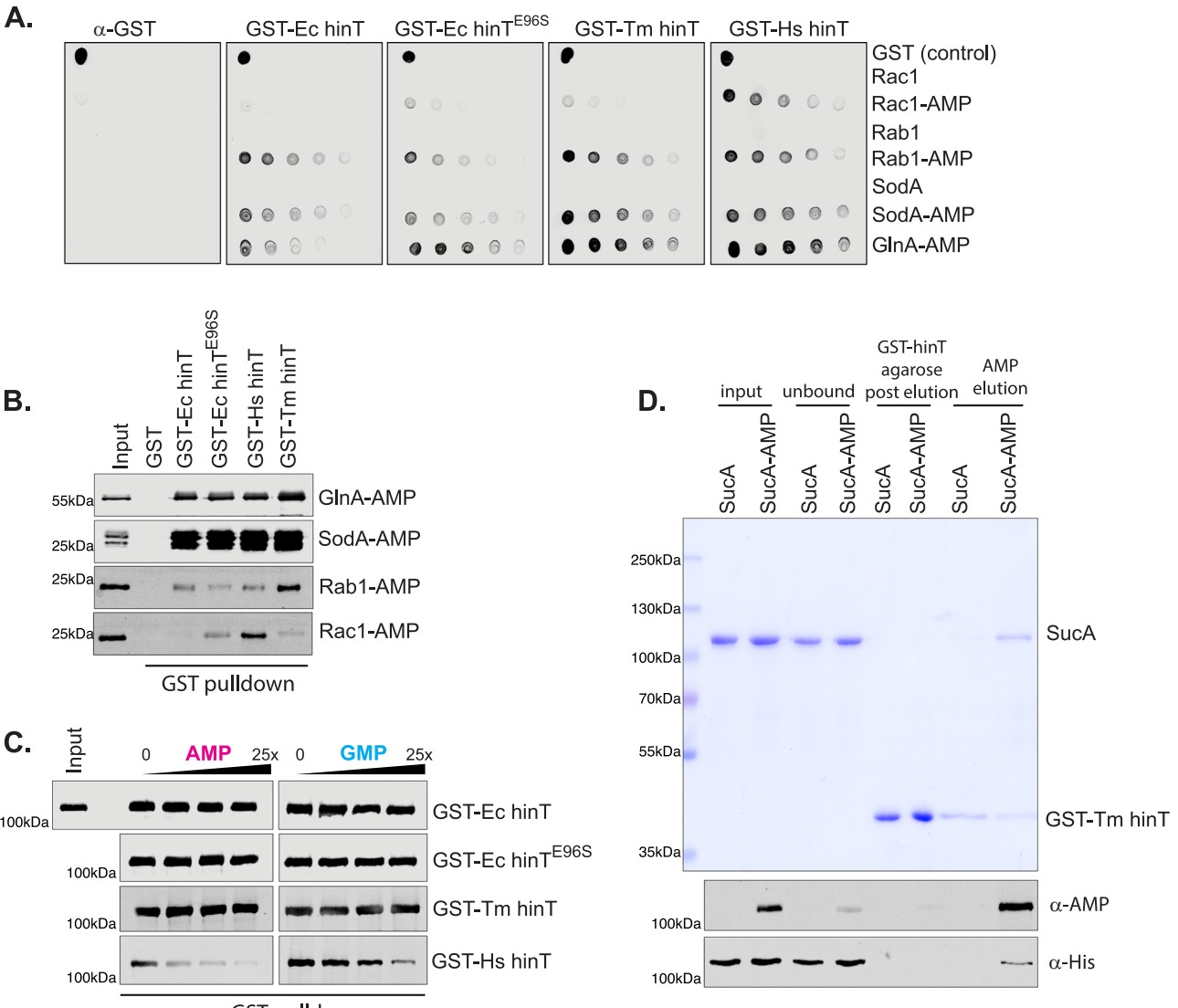

**Fig. 4 | hinT mutants display varying specificity and sensitivity to AMPylated substrates. A** Representative dot blot assay of serial dilutions (400 ng to 25 ng) of proteins spotted on nitrocellulose membranes and probed with GST, GST-Ec hinT, GST-Ec hinT^E96S, GST-Hs hinT or GST-Tm hinT. Bound proteins were visualized using α-GST immunoblotting. Results are representative of at least 3 independent experiments (Supplementary Fig. 9). **B** α-AMP immunoblotting of enriched proteins from pulldown assay of GST, GST-Ec hinT, GST-Ec hinT^E96S, GST-Hs hinT or GST-Tm hinT with AMPylated GlnA, SodA, Rab1 and Rac1. Results are representative

of at least 3 independent experiments (Supplementary Fig. 10b–e). **C** α-AMP immunoblotting of sucA-AMP bound to GST, GST-Ec hinT, GST-Ec hinT^E96S, GST-Hs hinT or GST-Tm hinT in the presence of increasing concentrations of AMP or GMP nucleotides. Results are representative of at least 3 independent experiments (Supplementary Fig. 11). **D** Coomassie blue staining analysis of AMPylated sucA purified using GST-Tm hinT affinity. Protein immunoblots for α-AMP and α-his are shown. Results are representative of at least 3 independent experiments. Source data are provided as a Source Data file.

GST-Ec hinT^HN, but not the GST control protein, bound to AMPylated Rab1, sodA and GlnA (Fig. 4A). Compared to Ec hinT^HN, mutation of E96S displayed increased binding to GlnA-AMP and Rac1-AMP. In addition, both *T. marianensis* hinT and *H. sapiens* hinT demonstrated enhanced sensitivity in binding to AMPylated proteins, but not to non-AMPylated proteins (Fig. 4A). In contrast to *E. coli* and *T. marianensis* hinT, human hinT bound Rac1-AMP as well as the other AMPylated proteins (Fig. 4A and Supplementary Fig. 9). Excitingly, the binding of human hinT^HN to threonine-AMPylated Rac1 indicates that the human homolog may possess altered binding specificity for AMPylated amino acids.

Next, we performed far-western blot analysis to analyze if hinT can detect denatured AMPylated proteins, similarly to AMPylation-specific antibodies (Supplementary Fig. 10a). In agreement with the dot blot analysis, Tm hinT^HN and Hs hinT^HN demonstrated increased binding to AMPylated proteins in comparison with Ec hinT^HN. Notably,

Rac1-AMP binding was only observed with Hs hinT^HN (Supplementary Fig. 10a). No binding was detected for BiP-AMP to hinT homologs in comparison to the α-AMP antibody (Supplementary Fig. 10a). We next analyzed the ability of hinT homologs to enrich AMPylated proteins using a pulldown assay (Fig. 4B and Supplementary Fig. 10b–e). GST tagged hinT^HN, but not GST alone, bound to GlnA-AMP and sodA-AMP. Notably, Rab1-AMP and Rac1-AMP were selectively enriched with Tm hinT^HN and Hs hinT^HN, respectively. Collectively, these studies demonstrate the sensitivity and specificity of hinT in the detection and isolation of AMPylated proteins.

Based on the affinity of hinT to AMP and published reports of purine nucleotide binding, we tested if AMPylated proteins can be competed with AMP or GMP nucleotides[27,28,40]. Ec hinT^HN and Tm hinT^HN displayed no change in binding to sucA-AMP even in the presence of a 25-fold molar excess of AMP or GMP (Fig. 4C and Supplementary Fig. 11). On the contrary, the binding of sucA-AMP to Hs hinT^HN

decreased with increasing concentration of AMP. Hs hinT[HN], which has the strongest affinity for AMP based on ITC, showed reduced binding at a 1:1 molar ratio of AMP nucleotide to AMPylated protein (Fig. 4C). In comparison to AMP, increasing concentrations of GMP showed weaker competition with AMPylated proteins for binding to Hs hinT[HN] (Fig. 4C).

Although AMP did not outcompete AMPylated proteins for binding to Tm hinT[HN] at the concentrations used in the competition assay, we reasoned that higher concentrations of AMP may compete with AMPylated proteins. Thus, we investigated if AMP competition could be used for elution in hinT-based affinity purification of recombinant AMPylated substrates. We purified His$_6$ tagged sucA or sucA-AMP by nickel-nitrilotriacetic acid (Ni-NTA) affinity chromatography and incubated eluted proteins with immobilized GST-Tm hinT[HN]. SucA-AMP, but not the non-AMPylated sucA, bound to hinT, and was effectively eluted using 100 mM AMP, as observed by protein immunoblotting using α-AMP and α-His antibodies (Fig. 4D). These findings, along with the BLI and ITC analysis, allow us to define specific roles for each of the hinT homologs, considering their respective advantages and limitations. Both Ec and Tm hinT[HN] are ideal for substrate enrichment, offering tight binding to AMPylated substrates and the ability to elute with a high concentration of AMP. In contrast, Hs hinT[HN] may be better suited for detection purposes, as it demonstrates greater sensitivity than commercial antibodies in recognizing certain AMPylated substrates (Supplementary Fig. 10a).

Substrate recognition by hinT relies heavily on its interactions with the AMP moiety. To test whether hinT exhibits cross-reactivity with AMP-like post-translational modifications, we investigated the potential interaction of hinT with ADP-ribosylated proteins (Supplementary Fig. 12). Protein ADP-ribosylation is a post-translational modification where ADP-ribose from NAD$^+$ is transferred to protein substrates by ADP-ribosyltransferase enzymes, such as poly ADP-ribose polymerase (PARP). In agreement with published reports, we detected oligo-ADP ribosylation and poly-ADP ribosylation of PARP1 using ADP-ribose-specific antibody[41]. In contrast, Tm hinT[HN] bound to sodA-AMP, but not ADP-ribosylated PARP1 (Supplementary Fig. 12a). To further test if hinT can detect ADP-ribosylated proteins in cell lysates, we performed far western blot analysis of cell lysates incubated with PARP1-HPF1 or PARP14. We observed robust protein ADP-ribosylation in lysates using the ADP-ribosylation specific antibodies, but not Tm hinT[HN] (Supplementary Fig. 12b). HinT shows little to no reactivity with ADP-ribosylated proteins at the tested concentrations but may exhibit residual binding toward highly abundant ADP-ribosylated proteins. Thus, hinT demonstrates high sensitivity and specificity for protein AMPylation, making it a powerful tool for advancing biochemical and cellular analysis.

### Identification of novel AMPylated proteins from mouse melanoma cell lines using hinT enrichment

SelO is an evolutionarily conserved pseudokinase that catalyzes protein AMPylation[33]. However, little is known about the eukaryotic substrates and functional importance of mammalian SelO. Our recent studies revealed a critical role for SelO in promoting melanoma metastasis[16]. Therefore, we used YUMM3.3 mouse melanoma cell lines as a model system to identify the substrates of SelO (Fig. 5A). The cellular signals that activate SelO for AMPylation, along with the enzymes responsible for removing the modification, remain unknown. To enhance the AMPylation signal, we generated YUMM3.3 SelO-deficient cell lines that stably express either SelO (YUMM3.3[SelO]) or the inactive D338A mutant (YUMM3.3[SelO D338A]), allowing us to test the effectiveness of the enrichment method.

Protein immunoblotting of cell lysates using an α-AMP antibody confirmed the presence of AMPylated proteins in cells expressing active SelO (Fig. 5B, **lane 1**). Non-specific or SelO-independent proteins that are reactive to α-AMP antibody were observed in lysates of cells

expressing inactive SelO D338A (Fig. 5B, **lane 2**). The only method currently available to enrich AMPylated proteins in cells, without the need for exogenous labeling, is immunoprecipitation with AMPylation-specific antibodies[20]. However, we did not detect any AMPylated proteins through immunoprecipitation using AMPylation antibodies (Fig. 5B, **lanes 3 and 4**). The prominent bands at 55 kDa and 25 kDa with the α-AMP immunoprecipitation are most likely the heavy and light chains of the antibodies used during immunoprecipitation. In contrast to immunoprecipitation, enrichment using GST-Ec hinT[HN] revealed several AMPylated proteins in cell extracts expressing active SelO, but not the inactive mutant (Fig. 5B, **lanes 5 and 6**). These studies highlight the significant advantage of using hinT as an enrichment method. The ability of hinT to selectively bind AMPylated substrates allows for greater sensitivity and specificity in detecting these modifications, even in complex cellular lysates.

To identify the enriched substrates, we performed label-free quantitative mass spectrometry on proteins enriched from YUMM3.3[SelO] or the inactive mutant YUMM3.3[SelO D338A] (Fig. 5C and Supplementary Data 2). Given the cellular localization of SelO, we filtered results for mitochondrial-associated proteins based on Uniprot ID mapping. We identified approximately 124 mitochondrial proteins that are significantly enriched in cells expressing SelO relative to the inactive mutant (fold change > 1.5 and $p$-value < 0.05, $n = 2$). Gene ontology analysis of the enriched proteins revealed that the top enriched KEGG pathways identified are the citrate cycle (TCA) and 2-oxocarboxylic acid metabolism (Fig. 5D). Furthermore, enriched proteins are linked to a variety of diseases, as these proteins play critical roles in maintaining cellular redox balance and metabolic pathways. These results demonstrate the utility of hinT as a novel and efficient enrichment method for the identification of AMPylated proteins in mammalian cells. Our mass spectrometry analysis identified novel substrates of SelO that regulate mitochondrial and cellular metabolism. A better understanding of mitochondrial protein AMPylation may uncover new facets of metabolic regulation which have important implications in both physiological and pathophysiological processes.

### SelO regulates cellular metabolism by AMPylation of glutamate dehydrogenase

To study the role of SelO in cellular metabolism, we assessed changes in metabolite levels in YUMM3.3[SelO] and YUMM3.3[SelO D338A]. Targeted metabolite analysis revealed 72 metabolites that are significantly altered in cells expressing SelO compared to the inactive mutant (Fig. 6A and Supplementary Fig. 13a). Among these metabolites, 15 are enriched while 57 are less abundant in YUMM3.3[SelO] than YUMM3.3[SelO D338A]. A principal component analysis of 273 metabolites demonstrated that cells expressing SelO are metabolically distinct from cells expressing inactive SelO (Supplementary Fig. 13b). Metabolite Set Enrichment Analysis (MSEA) revealed that metabolites related to glycolysis and pentose phosphate pathway were significantly reduced in YUMM3.3[SelO] in comparison to YUMM3.3[SelO D338A] (Fig. 6B).

The majority of proteins enriched from YUMM3.3[SelO] using hinT-based enrichment are metabolic enzymes (Fig. 5D). To determine if these proteins are genuine substrates of SelO and corroborate our proteomic and metabolomic data, we assessed AMPylation in cells expressing SelO, inactive mutant D348A, or the hyperactive mutant V334A using two distinct approaches. First, we co-expressed SelO with putative Flag-tagged substrates in HEK293a cells and assessed AMPylation of Flag immunoprecipitates using monoclonal α-AMP antibodies. SelO and the hyperactive V334A mutant, but not the inactive D338A, catalyzed AMPylation of multiple mitochondrial proteins (Supplementary Fig. 14a). Second, we used GST-Ec hinT[HN] to enrich for AMPylated proteins in YUMM3.3 cells and analyzed the enrichment by immunoblotting with the respective antibodies (Fig. 6C). GST-Ec hinT[HN] efficiently enriched several of the candidate proteins from

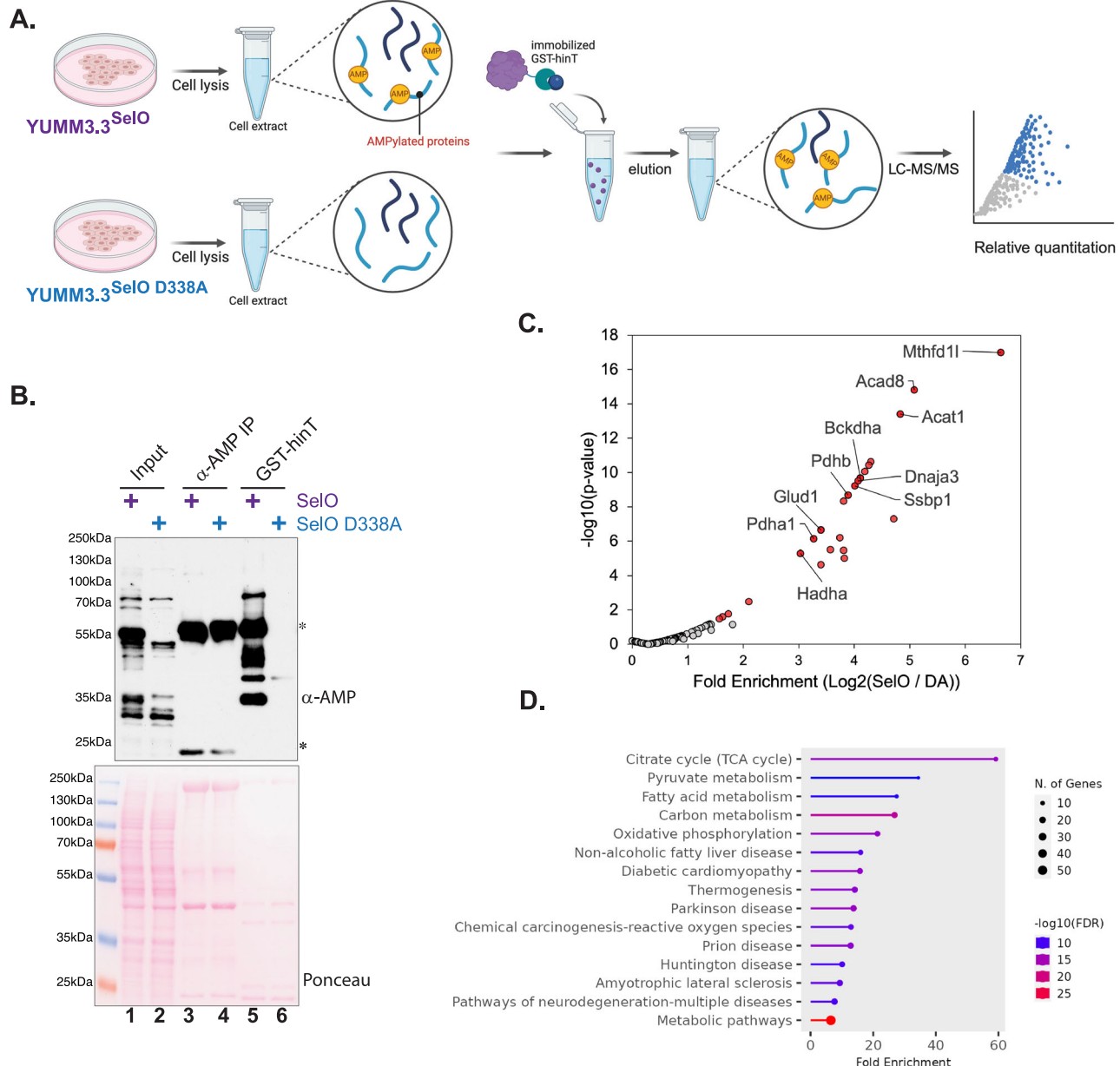

**Fig. 5 | Identification of novel AMPylated proteins from mouse melanoma cell lines using hinT. A** Schematic representation of the AMPylated protein enrichment strategy using hinT immobilized on glutathione agarose as bait to pulldown interacting AMPylated proteins from the melanoma cell line, YUMM3.3, stably expressing SelO or the inactive mutant D338A. Created in BioRender. Sreelatha, A. (2025) https://BioRender.com/h6gd6ci. **B** α-AMP immunoblotting of cell lysates, AMP immunoprecipitates, and GST-Ec hinT H101N enriched proteins from YUMM 3.3$^{SelO}$ or YUMM 3.3$^{D338A}$. Ponceau-stained membrane shows total protein staining. * denotes the positions of heavy and light chains of AMP antibodies used

for immunoprecipitation. Results are representative of 3 independent experiments. **C** Volcano plot depicting significance (-log10 *p*-value) versus Log2 (Abundance Ratio YUMM 3.3$^{SelO}$ / YUMM 3.3$^{D338A}$) of all mitochondrial-associated proteins identified by LC-MS/MS of hinT enriched proteins depicted in lanes 5 and 6 from (**B**). Proteins in red display > 1.5-fold change in YUMM 3.3$^{SelO}$ / YUMM 3.3$^{D338A}$ and *p*-values < 0.05. The top 10 most abundant mitochondrial proteins (highest Sum PEP Score) are labeled. *P*-values were calculated using an ANOVA test. **D** Enriched GO biological process terms for mitochondrial proteins enriched in YUMM 3.3$^{SelO}$ in comparison to YUMM 3.3$^{D338A}$ from (**C**).

lysates of cells expressing SelO and the hyperactive mutant, but not the inactive mutant.

Among our top candidates is Glud1, glutamate dehydrogenase and PdhB, β subunit of pyruvate dehydrogenase E1 complex. Notably, we observed a mobility shift in Glud1 from cell lysates expressing SelO and the hyperactive mutant, but not the inactive mutant (Fig. 6C). The higher molecular weight species of Glud1 was selectively enriched using hinT, and its signal overlapped with robust AMPylation, as detected by α-AMP antibody. To further confirm that Glud1 AMPylation is dependent on the catalytic activity of SelO, we expressed Flag-tagged Glud1 and analyzed Flag immunoprecipitates

from YUMM3.3$^{SelO}$ and YUMM3.3$^{SelO\ D338A}$ for AMPylation. Protein immunoblotting using α-AMP antibody confirmed AMPylation of Glud1-Flag isolated from YUMM3.3$^{SelO}$ but not YUMM3.3$^{SelO\ D338A}$ lysates (Fig. 6D and Supplementary Fig. 14b, c). To confirm and identify the AMPylation site on Glud1, we performed MS analysis on Flag immunoprecipitates from YUMM3.3$^{SelO}$ and YUMM3.3$^{SelO\ D338A}$ lysates. We identified AMPylated tryptic peptides of Glud1 that were present in YUMM3.3$^{SelO}$ but not YUMM3.3$^{SelO\ D338A}$ (Fig. 6E and Supplementary Fig. 15a). Notably, we identified Y464 which is located in the regulatory antennae domain of Glud1 as a potential site of modification (Supplementary Fig. 15b).

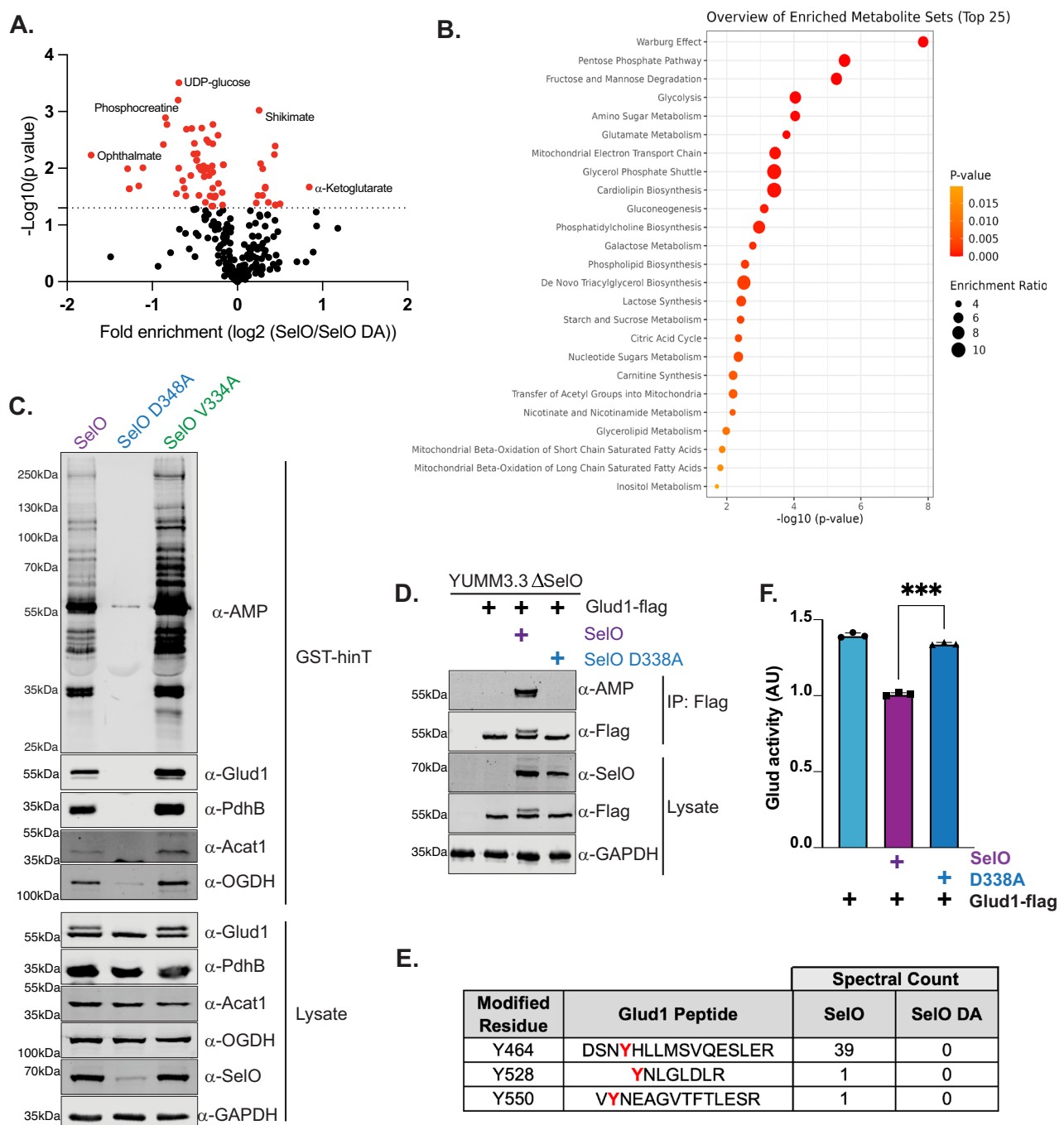

**Fig. 6 | SelO-mediated AMPylation inactivates glutamate dehydrogenase and alters cellular metabolism. A** Volcano plot displaying metabolite abundance in whole cell extract of YUMM 3.3[SelO] and YUMM 3.3[D338A] obtained by targeted LC-MS. Metabolites in red indicate *p*-value < 0.05. *P*-values were calculated using a two-tailed *t* test assuming unequal variances. **B** Metabolite set enrichment analysis (MSEA) of metabolites significantly reduced in YUMM 3.3[SelO] in comparison to YUMM 3.3[D338A] from (**C**). Analysis was performed using MetaboAnalyst 6.0. One-tailed *p* values are provided after adjusting for multiple testing. **C** Protein immunoblotting of GST-Ec hinT H101N enriched proteins and input cell lysates from YUMM 3.3[SelO], YUMM 3.3[D348A], or YUMM 3.3[V334A]. Results are representative of 3 independent experiments. **D** Protein immunoblotting of flag immunoprecipitates

or cell lysates from YUMM3.3ΔSelO expressing Glud1-flag or Glud1-flag and SelO or SelO[D338A]. Results are representative of at least 3 independent experiments (Supplementary Fig. 14b, c). **E** AMPylated peptides identified by MS/MS analysis of Glud1 from flag immunoprecipitates of YUMM3.3 co-expressing Glud1-flag with either SelO or SelO[D338A]. AMPylation sites are highlighted in red. Spectral counts reflect the number of MS2 peptide spectral matches as determined by Mascot software searches. **F** The activity of Glud1 in YUMM3.3ΔSelO expressing Glud1-flag or Glud1-flag and SelO or SelO[D338A]. Data represent the average of 3 technical replicates. Significance calculated by two-tailed *t* test, *p* = 0.0003. Data are presented as mean values +/− SD, *n* = 3. Results are representative of at least 3 biological replicates. Source data are provided as a Source Data file.

Glutamate dehydrogenase catalyzes the deamination of glutamate into 2-oxoglutarate and ammonia. To test the effect of SelO-mediated AMPylation on Glud1, we assessed glutamate dehydrogenase activity and observed that Glud1 from YUMM3.3[SelO] had

significantly reduced activity in comparison to YUMM3.3[SelO D338A] (Fig. 6F). Collectively, these results suggest that SelO inactivates Glud1 by AMPylation of the regulatory antennae to alter cellular metabolism.

## Discussion

Protein AMPylation was initially discovered as a regulatory mechanism in bacterial nitrogen metabolism. Recent studies have revealed a role for AMPylation in bacterial infection, the mammalian unfolded protein response and mitochondrial antioxidant signaling. Although chemical proteomic methods using ATP analogs have facilitated the enrichment of AMPylated proteins, there are several disadvantages, including the need for in vitro labeling. The reduced intracellular uptake and sensitivity of these probes limit their application in cells and hinder the identification of novel AMPylated substrates. Hence, developing tools to study AMPylation is essential to advance our understanding of this emerging PTM.

We developed a novel strategy using a modified nucleotide-binding protein, hinT, which offers several advantages over previously established methods for AMP detection. Notably, it allows for unbiased profiling without the need for exogenous ATP analogs or recombinant AMPylases, and therefore limits artifacts arising from suboptimal labeling conditions and enzyme concentrations in vitro. In comparison to AMPylation specific antibodies, our method utilizes recombinant proteins that can be easily generated in *E. coli* and further optimized for sensitivity and specificity. The structure of AMPylated GlnA bound to hinT provides a platform for protein engineering to optimize the binding interface to alter specificity and sensitivity for AMPylated proteins.

The structure of hinT bound to AMPylated GlnA depicts the AMP moiety of Tyr-AMP bound in the nucleotide binding pocket with unique interactions that precisely coordinate the phosphate and purine ring. We screened homologs and performed mutational analysis on hinT residues which interact with the AMPylated tyrosine in GlnA to optimize substrate specificity and affinity. The *H. sapiens* and *T. marianensis* homologs not only show improved binding but also differential specificity for threonine or tyrosine AMPylation. Importantly, the hinT proteins did not bind to other PTMs such as ADP-ribosylation. Comparison of the binding pocket of hinT with other nucleotide-binding proteins, such as the AMP-binding Cystathionine Beta Synthase (CBS) domain, reveals that hinT binds AMP in the proper orientation and spatial arrangement. While hinT exhibits specific AMP binding, its more accessible binding site provides ample space for the AMPylated protein. This characteristic makes hinT a more promising candidate for specifically binding AMP moieties in AMPylated proteins (Supplementary Fig. 16). Studies are underway to improve the binding of hinT to AMPylated substrates with the use of more thorough mutational and computational analysis.

We show that GST-hinT[HN] binds AMPylated substrates from cell lysates more efficiently than commercial antibodies for AMPylation. In addition, this versatile system can be adapted for various biochemical techniques, including pull-down assays, far-western blots, and the purification of AMPylated substrates for activity assays. AMP or purine base nucleotides can be used to elute AMPylated substrates from hinT by competition. Although competitive elution offers a distinct advantage in the purification of AMPylated proteins, it can also be a limiting factor in the isolation of substrates from cells due to the high concentration of cellular nucleotides. To optimize binding of AMPylated proteins, charcoal columns or enzymatic digestion can be used to deplete cellular nucleotides. In addition, hinT may interact with nucleotide-binding proteins, including RNA-binding proteins, but this can be effectively filtered out using an AMPylation deficient control. Despite our current limitations, we identified several AMPylated proteins that are enriched with GST-hinT[HN] in cells expressing SelO, but not the inactive SelO mutant. These findings allow us to explore the function of protein AMPylation in the mitochondria. Thus, our studies describe a promising new tool for the detection, identification, and purification of an array of AMPylated substrates.

To gain an understanding of the functional importance of the evolutionarily conserved AMPylase SelO, we used GST-hinT[HN] to enrich for SelO-mediated AMPylated proteins from YUMM3.3 melanoma cell lines. We identified several substrates that are part of the metabolic pathways in the mitochondria. One such AMPylated protein is glutamate dehydrogenase, which is AMPylated and less active in cells expressing SelO, but not SelO[D338A]. Mass spectrometry analysis revealed the putative site of AMPylation as Y464 in the antenna domain of Glud1 (Supplementary Fig. 17). The antenna plays an important role in allosteric regulation for both inhibition and activation of the enzyme by different cofactors. AMPylation of the tyrosine residue on Glud1's antenna may be an additional mechanism for enzyme inhibition to maintain cellular energy balance and ammonia levels. Analogous to the regulation of nitrogen metabolism in bacteria through the AMPylation of GlnA, Glud1 AMPylation could have a similar regulatory function for nitrogen metabolism in eukaryotes. Future work will evaluate the physiological importance of Glud1 AMPylation in the mitochondria.

AMPylation is emerging as a prominent PTM in mammalian cells, and our knowledge of this PTM is in its infancy. Currently, there are only a handful of enzymes known to catalyze AMPylation. There may be many more AMPylases and AMPylated substrates that have evaded detection due to the low abundance and stoichiometry of AMPylation. Akin to SH2 binding domains, which have been valuable tools for the analysis of protein tyrosine phosphorylation, our strategy to isolate endogenous AMPylated proteins from cell lysates may enhance our understanding of the fundamental mechanisms and the functional importance of AMPylation in cell signaling.

## Methods

### Reagents and bacterial strains

Glutathione agarose was purchase from Fisher scientific (PI16101). Mouse α-AMP antibodies (clone IDs B992601, B992602, and B992603) were purchased from Biointron based on ref. 20. *E. coli* BW25113 and knockout strains of GlnA (JW3841-1), GlnE (JW3025-1), SelO (JW1696-1) were obtained from the *E. coli* genetic stock center (CGSC) at Yale.

### Generation of constructs

*E. coli* SelO ppSumo and *E. coli* SelO D348A ppSumo were generated in ref. 15. The *E. coli* hinT coding sequence was amplified by PCR using DH5α genomic DNA as template. The open reading frame was cloned into bacterial expression vectors, pProEx, or a modified pGEX-rTEV, which contains a glutathione S-transferase (GST) tag followed by a TEV cleavage site. The *H. sapiens* hinT coding sequence was amplified from the Ultimate ORF Lite human cDNA collection (Life Technologies NM_005340.6). *T. marianensis* was synthesized as a gBlock gene fragment (IDT). *E. coli* SelO V242A ppSumo and E96S Ec hinT pGex-rTEV were generated by site directed mutagenesis using Agilent Pfu-Turbo DNA polymerase (Supplementary Table 3).

### Protein expression and purification

BL21 cells were transformed with hinT (*E.coli* H101N, *H. sapiens* H112N, *T. marianensis* H100N) pProEx and grown in Luria-Bertani (LB) media containing 100 μg/ml of ampicillin at 37 °C. Cells were then induced at an approximate $OD_{600}$ of 0.6 with 0.4 mM IPTG and grown for 16–18 h at 22 °C. Cells were harvested by centrifugation and resuspended in lysis buffer (25 mM Tris, pH 8, 10 mM imidazole pH 8, 300 mM NaCl, 10% Glycerol, 1 mM PMSF and 1 mM DTT) and lysed by sonication. Lysate was clarified by centrifugation at 25000 x $g$ for 25 min, and supernatant was incubated with Ni-NTA beads for one hour at 4 °C. Samples were passed over a column, and beads were washed with 20 CV of lysis buffer. Proteins were eluted with 5 CV of 25 mM Tris pH 8, 300 mM imidazole pH 8, 150 mM NaCl, 10% Glycerol, and 1 mM DTT. 6xHis tag was then cleaved by incubating eluted proteins with TEV protease overnight at 4 °C. hinT was further purified by size exclusion chromatography using a HiLoad 16/600 Superdex 75 attached to a BioRad NGC chromatography system pre-equilibrated with 10 mM Tris-HCl, pH 8, 150 mM NaCl, and 2 mM DTT.

*H. sapiens* hinT H112N recombinant proteins used for ITC was denatured and refolded to decrease copurifying AMP as previously described[30]. Briefly, fractions collected from Superdex 75 were concentrated and dialyzed overnight against 8 M Urea. The sample was sequentially dialyzed against buffer with 4 M Urea, 2 M Urea, then TBS (10 mM Tris-HCl, pH 8, 150 mM NaCl and 2 mM DTT) without urea at 4 °C for refolding.

BL21 cells were transformed with hinT (*E.coli* H101N, *H. sapiens* H112N, *T. marianensis* H100N) pGEX-rTEV and grown in Luria-Bertani (LB) media containing 100 µg/ml of ampicillin at 37 °C. Cells were then induced at an approximate $OD_{600}$ of 0.6 with 0.4 mM IPTG and grown for 16–18 h at 22 °C. Cells were harvested by centrifugation and resuspended in lysis buffer (25 mM Tris pH 8, 300 mM NaCl, 10% Glycerol, 1 mM PMSF and 1 mM DTT) and lysed by sonication. Lysate was clarified by centrifugation at 25000 x *g* for 25 min, and supernatant was incubated with glutathione agarose beads for one hour at 4 °C. Samples were passed over a column, and beads were washed with 20 CV of lysis buffer. Proteins were eluted with 5 CV of 25 mM Tris pH 8, 20 mM glutathione, 150 mM NaCl, 10% Glycerol, and 1 mM DTT. hinT was further purified by size exclusion chromatography using a HiLoad 16/600 Superdex 75 pre-equilibrated with 10 mM Tris-HCl, pH 8, 150 mM NaCl containing 2 mM DTT.

BL21 cells were transformed with GlnA ppSumo and grown in LB media containing 50 µg/ml of kanamycin at 37 °C. Cells were then induced at an $OD_{600}$ of 0.6 with 0.4 mM IPTG and grown for 16–18 h at 22 °C. Cells were then harvested by centrifugation and resuspended in 25 mM Tris, pH 8, 10 mM imidazole, pH 8, 300 mM NaCl, 10% Glycerol, and 1 mM DTT. Cells were lysed by sonication and clarified by centrifugation at 25000 x *g* for 25 min. Supernatant was incubated with Ni-NTA beads for one hour at 4 °C. Samples were passed over a column, and beads were washed with 20 CV of lysis buffer. Proteins were eluted with 5 CV of 25 mM Tris pH 8, 300 mM imidazole pH 8, 300 mM NaCl, 10% Glycerol, and 1 mM DTT. Eluted proteins were incubated with immobilized GST-hinT H101N for 1 h at 4 °C for AMPylated GlnA enrichment. Beads containing the immobilized GST-hinT H101N were then passed over a column and washed with 5 CV lysis buffer supplemented with 10 mM MnCl2 to remove unbound GlnA. AMPylated GlnA was eluted with lysis buffer supplemented with 100 mM AMP. 6xHis-SUMO tag was then removed by cleavage with ULP1 at 4 °C overnight. GlnA was further purified by size exclusion chromatography using a HiLoad 16/600 Superdex 200 column attached to a BioRad NGC chromatography system pre-equilibrated with 10 mM Tris-HCl, pH 8, 150 mM NaCl, and 2 mM DTT.

AMPylated sodA was generated by co-expressing *E. coli* SelO ppSUMO and sodA pProEX in Rosetta DE3 cells. Cells were grown in LB broth supplemented with 100 µg/mL ampicillin and 50 µg/mL kanamycin to an $OD_{600}$ of 0.6–0.8. Protein expression was induced with 0.4 mM IPTG for 16–18 h at 22 °C. Cells were harvested by centrifugation and lysed in 50 mM Tris- HCl pH 8, 300 mM NaCl, 1 mM PMSF and 1 mM DTT by sonication. Lysate was clarified by centrifugation at 25,000 × *g* for 25 min. The soluble lysate was incubated with Ni-NTA beads for one hour at 4 °C. Beads were passed through a column and washed with 20 column volumes of 50 mM Tris-HCl, pH 8, 300 mM NaCl, 10 mM imidazole and 1 mM DTT. The protein was eluted with 50 mM Tris-HCl, pH 8, 300 mM NaCl, 300 mM imidazole, 1 mM DTT. The eluted protein was then buffer exchanged to 25 mM Tris, pH 7.5, 1 mM DTT. The AMPylated sodA protein was further purified using an EnrichQ ion exchange column attached to a BioRad NGC chromatography system.

For unmodified or AMPylated sucA expression, BL21 cells were transformed with sucA:SelO D256A petDuet[15] or sucA:SelO V242A pETDuet, and grown in LB media containing 100 µg/ml of ampicillin at 37 °C. Cells were then induced at an $OD_{600}$ of 0.6 with 0.4 mM IPTG and grown for 16–18 h at 22 °C. Cells were then harvested by centrifugation and resuspended in lysis buffer (25 mM Tris, pH 8, 10 mM

imidazole pH 8, 300 mM NaCl, 10% Glycerol, 1 mM PMSF, and 1 mM DTT). Cells were lysed by sonication and clarified by centrifugation at 25000 x *g* for 25 min. Supernatant was incubated with Ni-NTA beads for one hour at 4 °C. Samples were passed over a column, and beads were washed with 20 CV of lysis buffer. Proteins were eluted with 5 CV of 25 mM Tris pH 8, 300 mM imidazole pH 8, 300 mM NaCl, 10% Glycerol, and 1 mM DTT.

Unmodified sucA was further purified by size exclusion using HiLoad 16/600 Superdex 200 column attached to a BioRad NGC chromatography system pre-equilibrated with 10 mM Tris-HCl, pH 8, 150 mM NaCl, and 2 mM DTT. AMPylated sucA was buffer exchanged to 25 mM Tris pH 8, 300 mM NaCl, 10% Glycerol, 1 mM DTT by centrifugation and incubated with immobilized GST-hinT H101N overnight at 4 °C for AMPylated sucA enrichment. Beads containing the immobilized GST-hinT H101N were then passed over a column and washed with 5 CV lysis buffer to remove unbound sucA. AMPylated sucA was eluted with 25 mM Tris, pH 8, 150 mM NaCl, 100 mM AMP, 2 mM DTT. SucA was further purified by size exclusion chromatography using a Superose 6 column attached to a BioRad NGC chromatography system pre-equilibrated with 10 mM Tris-HCl, pH 8, 150 mM NaCl, and 2 mM DTT.

## BiP AMPylation assay
BiP and FicD E234G recombinant proteins were gifted by the Orth lab. AMPylation assay was performed in 25 mM HEPES pH 7.5, 100 mM KCl, 5 mM MgCl2, 2 mM DTT, 250 µM ATP, 0.05% TX-100 with 4 µM BiP and 133 nM FicD E234G. Reaction was incubated at 37 °C for 30 min and stopped with SDS loading buffer containing 1% β-mercaptoethanol.

## ADP ribosylation assay
Purified PARP1, PARP14 and HPF recombinant proteins were gifted by the Liszczak lab[42]. ADP ribosylation assay reactions were performed on 40 µL total volume of ADP-ribosylation buffer containing 50 mM Tris, pH 7.5, 20 mM NaCl, 2 mM MgCl2 and 5 mM DTT. Each reaction contained 150 µg HEK293a lysates and either no enzyme for negative control, 1 µM PARP1 and 25 µM HPF, or 100 µM PARP14. All reactions were supplemented with 1 mM $NAD^+$ and incubated at 37 °C for 30 min. Reactions were stopped by the addition of SDS loading buffer with 1% β-mercaptoethanol.

Auto ADP ribosylation assay of PARP1 was performed in 50 mM Tris pH 7.5, 20 mM NaCl, 2 mM MgCl2 and 5 mM DTT with 80 ng/µL PARP1. Each reaction was supplemented with either no $NAD^+$, 3 µM $NAD^+$ or 250 µM $NAD^+$ and incubated at 37 °C for 30 min. Reaction was stopped by the addition of 5 mM EDTA.

## Cryo-EM sample preparation and data acquisition
For the GlnA-hint complex, 8 µl of purified GlnA (4.68 mg/ml) and 9 µl of purified hinT (3.29 mg/ml) in 10 mM Tris, pH 8, 150 mM NaCl, 2 mM DTT were mixed for a molar ratio of 1:3 molar ratio in monomer consideration, and around 1:18 GlnA dodecamer to hinT dimer consideration. GlnA-hinT complex was loaded onto 300-mesh copper grids (Quantifoil R1.2/1.3 300-mesh copper grids were glow-discharged using a PELCO easiGlow glow discharge apparatus at 30 mA/30 s on top of a Ted Pella metal grid holder). GlnA-hinT excess sample was blotted for 3 s before plunge-freezing in a ThermoFisher Vitrobot System at 4 °C and 95% humidity.

Cryo-EM data were collected using a Titan Krios microscope equipped with the post-column BioQuantum energy filter (Gatan) connected to a K3 direct electron detector (Gatan). Cryo-EM data were collected using SerialEM[43] in a super-resolution counting mode with a 20 eV energy filter slit. A total exposure time of 4.6 s with 77 ms per frame resulted in a 60-frame movie per exposure with an accumulated dose of 60 e-/Å². The calibrated physical pixel size and the super-resolution pixel size are 0.83 Å and 0.415 Å, respectively. The defocus in data collection was set in the range of − 0.9 to

− 2.2 µm. A total of 5758 movies were collected and used for data processing.

## Cryo-EM data processing

Cryo-EM data were processed using Relion 4.0.1[44–46]. Beam induced motion-correction and dose-weighting to compensate for radiation damage over spatial frequencies were perform using MotionCor2[47] with a binning factor of 2, resulting in a pixel size of 0.83 Å/pixel for the micrographs. Contrast Transfer Function (CTF) parameters were estimated using GCTF[48]. 5683 micrographs were selected for further processing from the total of 5758 after manual curation. Particles were selected using Gautomatch (K. Zhang, MRC LMB, https://www2.mrc-lmb.cam.ac.uk/download/gautomatch-053/). A total of 989,142 particles were extracted. Next, particles were subjected to one round of 2D classification in Relion, followed by one round of 3D classification with alignment on selected 609,241 particles from the 2D classification job. No symmetry was imposed at this step. A subset of the particles selected from the 2D classification job were used to generate the ab initio model in Relion, which was used in the 3D classification job as the initial reference. 495,703 particles were selected after 3D classification and were further refined using Relion with C6 symmetry. Beam tilt, anisotropic magnification, and per-particle CTF estimations were performed in Relion to improve the resolution of the final reconstruction of the GlnA dodecamer complex, resulting in a density map with an overall resolution of 2.2 Å, which was further sharpened by DeepEMhancer[49]. All resolution was reported according to the gold-standard Fourier shell correlation (FSC) using the 0.143 criterion[50]. Local resolution was estimated using Relion.

To gain structural insight into hinT binding to the AMPylated GlnA, we performed further data processing in Relion. First, the 495,703 particles corresponding to the dodecamer were expanded according to the 6-fold symmetry. This resulted in a total of 2,974,218 particles after symmetry expansion. Although there are 12 copies of GlnA in the complex, there are only 6 possible hinT binding sites present. Next, a soft mask was generated. This mask centered on one hinT binding site, covering the hinT density as well as two adjacent GlnA molecules, one from each layer of the dodecamer (Supplementary Fig. 3). This mask was then applied in a 3D classification job without particle alignment, from which a total of 1,139,875 particles with clear hinT density were selected. Other classes in this classification had no or very weak density for hinT, indicating that not all the sites were occupied by hinT. Next, signal subtraction was performed to keep only the signal within the soft mask. The resulted particles were then re-centered according to the 2-fold symmetric axis between the two copies of GlnA and re-refined with C2 symmetry. Another soft mask covering only the hinT density was then applied in a subsequent 3D classification job without particle alignment. From this classification, two classes were identified with a clear feature of the hinT dimer. These two classes could be related by a 180° rotation, and therefore were combined after the rotational operation on one of the classes, resulting in 376,052 particles (Supplementary Fig. 3). Subsequent refinement was done without any symmetry operation. Next, another round of 3D classification without particle alignment was performed with a soft mask around the hinT dimer region (Supplementary Fig. 3). The classes with good features of hinT dimer were selected (225,422 particles) and were further refined to 2.7 Å overall resolution. In this map, the density for the hinT dimer was less well resolved. Therefore, a local refinement with a hinT mask was performed to improve the map quality of the hinT dimer. This resulted in a 3.5 Å overall resolution map (Supplementary Fig. 3). Both maps were further sharpened by DeepEMhancer[49]. The locally refined map also contained density from GlnA; the soft mask used for the local refinement was applied again to the DeepEMhancer sharpened map to generate a map only containing the hinT dimer plus the loops from GlnA. This map was combined with the DeepEMhancer sharpened map prior to local refinement to generate a composite map using the "*vop maximum*" function in UCSF Chimera based on the maximum value at each voxel[51]. This composite map is used to show the features of hinT H101N binding to the AMPylated GlnA.

## Atomic model building

GlnA dodecamer and GlnA dimer models were extracted from the previous structure GlnA structure (PDB: 7W85). hinT dimer was extracted from the previous structure (PDB:3N1T). Extracted models were then fitted to maps in Chimera X and utilized as initial models for model building using ISOLDE[52] in UCSF Chimera X[51] and COOT[53] against the DeepEMhancer sharpened maps. The model was built through iterations of real-space refinement in Phenix[54] with secondary structure restraints. Model geometries were assessed using the MolProbity module in Phenix, the MolProbity server[55] (http://molprobity.biochem.duke.edu/), and the PDB Validation server[56] (www.wwpdb.org). Figures were prepared using UCSF ChimeraX[51].

## Mass spectrometry analysis

Protein samples were resolved on SDS-PAGE gels and stained with Coomassie blue dye prior to mass spectrometry analysis. Proteins were first reduced with 10 mM DTT for 1 hr at 56 °C and then alkylated with 50 mM iodoacetamide for 45 min at room temperature in the dark. Overnight enzymatic digestion with trypsin (MS grade) was carried out at 37 °C. Resulting tryptic peptides were de-salted via solid phase extraction (SPE) before analysis. LC-MS/MS experiments were performed on a Thermo Scientific EASY-nLC liquid chromatography system coupled to a Thermo Orbitrap Fusion Lumos mass spectrometer. To generate MS/MS spectra, MS1 spectra were acquired in the Orbitrap mass analyzer (resolution 120,000), and peptide precursor ions were then isolated and fragmented using high-energy collision-induced dissociation (HCD). The resulting MS/MS fragmentation spectra were acquired in the ion trap. MS/MS spectral data was searched using Proteome Discoverer 2.2 software (Thermo) or the Mascot search engine (Matrix Science) against sequences in the Uniprot Escherichia coli (strain K12) protein database (Taxon ID 83333) or the Uniprot Mus musculus protein database (Taxon ID 10090). For all searches, the precursor mass tolerance was 15 ppm and the fragment mass tolerance was 0.6 Da. Peptide spectral matches were adjusted to a 1% false discovery rate (FDR), and additionally proteins were filtered to a 5% FDR. Variable modifications included Carbamidomethylation of cysteine (+ 57.021 Da), oxidation of methionine (+ 15.995 Da), and acetylation of protein N-termini (+ 42.011 Da). For the identification of AMPylated peptides (Fig. 6E, Supplementary Figs. 1A, 15a), Phosphoadenosine (+329.053 Da) was also included as a variable modification on serine/tyrosine/threonine. AMPylation sites were manually verified from MS/MS spectral data.

Supplementary Table 1 shows only the Top 10 highest-scoring scoring *E.coli* proteins identified from the 55 kDa band observed in the GST-hinT[HN] *E.coli* lysate pulldown shown in Fig. 1. Supplementary Data 1 and 2 show results from label-free quantitative analysis using Proteome Discoverer software. Protein abundance values are based on calculated areas of precursor ions, and ratios were calculated using a pairwise ratio-based approach. Enriched proteins required an abundance ratio fold change > 1.5, and proteins only found in SelO but not SelO DA were assigned the max Abundance Ratio of 100. Smear plots for Supplementary Fig. 1b, c and volcano plots for Fig. 5C were generated from their respective label-free quantitative datasets shown in Supplementary Data 1 or Supplementary Data 2.

Common contaminant proteins, such as keratins, that are listed in the Global Proteome Machine Organization's Common Repository of Adventitious Proteins were removed from all lists. For YUMM3.3 mouse melanoma cell lines, mitochondrial proteins were assigned based on UniprotKB retrieve/ID mapping feature (https://www.uniprot.org/uploadlists/) to identify proteins with a mitochondrial

subcellular location or Gene Ontology cellular component GO term. Enrichment analysis of the set of mitochondrial-associated genes was performed using ShinyGO (http://bioinformatics.sdstate.edu/go/) with an FDR cutoff of 0.05 applied[57].

### Isothermal titration calorimetry

hinT protein samples were dialyzed against 10 mM Tris pH 8, 150 mM NaCl, 2 mM β-mercaptoethanol. ITC experiments were performed in a Micro-Cal PEAQ-ITC (Malvern Panalytical, Worcestershire, UK) calorimeter with a stirred 206.2 μL reaction cell held at 20 °C. The first injections were 0.5 μL, followed by twenty 1.9 μL injections with a stirring rate of 750 rpm. *E.coli* hinT was used at 100 μM in the ITC cell; the syringe was filled with 1 mM AMP for titration. *H. sapiens* hinT was used at 30 μM in the ITC; 300 μM AMP was used in the syringe. *T. marianensis* hinT was used at 50 μM in the cell; 500 μM AMP was used on the syringe. All ITC experiments were performed in duplicate. ITC data were integrated and baseline corrected using NITPIC[58]. The integrated data were globally analyzed in SEDPHAT[59] using a model considering a single class of binding sites. Thermogram and binding figures were plotted in GUSSI[60].

### Biolayer interferometry

BLI experiments were performed in a Sartorius Octet R8 instrument. GST-tagged hinT, sucA, and sucA-AMP recombinant proteins were diluted in 10 mM Tris, pH 8, 150 mM NaCl, 0.1% Triton X-100 (TBSTX-100). GST-tagged hinT recombinant proteins were immobilized to Octet anti-GST biosensors (Sartorius 18-5096). Pins were equilibrated in TBSTX-100 for 10 minutes, followed by a 3-minute loading step with GST-hinT at 90 nM. Pins were then submerged on TBSTX-100 to wash unbound samples, followed by the sucA-AMP binding step and disassociation step for 5 minutes each. Binding data was collected for serial dilutions of sucA-AMP from 500 nM, 250 nM, 125 nM, 62.5 nM, 31.3 nM, 15.6 nM, 7.81 nM and buffer only control. For unmodified sucA, binding data was collected at the highest concentration of 500 nM. Raw and steady state data values were extracted from Octet® Analysis Studio and plotted using GraphPad Prism. Data was processed using a double reference subtraction using the signal from immobilized GST-hinT proteins dipping into buffer (no sucA-AMP control) and GST-loaded pins dipping into each concentration of sucA-AMP. $K_D$ values were determined from steady-state binding responses as a non-linear regression curve with one site-specific binding.

### hinT pulldown of recombinant AMPylated substrates

5 μg of hinT recombinant proteins were immobilized to 10 μL of glutathione agarose beads for 1 h. Beads were then washed 3 times and resuspended with TBSTX-100 containing 1 mM DTT. 100 μL of beads resuspension was incubated with 2 μg of AMPylated substrates and nutated for 1 h at 4 °C. Beads were centrifuged 5000 × *g* for 30 s and washed three times with TBSTX-100. Beads were resuspended in SDS loading buffer containing 1% β-mercaptoethanol and boiled. Samples were resolved by SDS-PAGE, transferred to nitrocellulose membranes and analyzed for protein AMPylation using 0.2 μg/ml α-AMPylation (clone 1G11) and 0.34 μg/ml α-AMPylation (clone 17G6) antibodies.

### Purine nucleotide competition assay

For the purine nucleotide competition assay, 5 μg of hinT was immobilized to 10 μL of glutathione agarose beads for 1 h. Beads were then washed 3 times and resuspended with TBSTX-100 containing 1 mM DTT. 100 μL of beads resuspension was incubated with 5 μg of sucA-AMP (470 nM) and either no nucleotide, 1:1 molar ratio nucleotide (470 nM), 1:5 molar ratio nucleotide (2.35 μM), or 1:25 molar ratio nucleotide (11.75 μM). Binding reactions were incubated for 1 h at 4 °C. Beads were centrifuged 5000 × *g* for 30 s and washed three times with TBSTX-100. Input, unbound, post-elution beads, and eluted proteins

were resuspended in SDS loading buffer containing 1% β-mercaptoethanol and boiled. Samples were resolved by SDS-PAGE, transferred to nitrocellulose membranes and analyzed for protein AMPylation using 0.13 μg/ml α-AMPylation (clone 1G11) antibodies.

### Purification of AMPylated sucA

25 μg of GST-hinT was immobilized to 10 μL of glutathione agarose beads for 2 h at 4 °C. Beads were washed 3 times with 500 μL TBSTX-100. 25 μg of Ni-NTA purified non-AMPylated sucA or sucA-AMP was added to beads and incubated overnight at 4 °C. Beads were then spun down and washed 3 times with 500 μL TBSTX-100 buffer. Samples were eluted by incubating beads with 20 μL TBSTX-100 containing 100 mM AMP for 10 min, 4 times. Samples were resuspended in SDS loading buffer containing 1% β-mercaptoethanol and boiled. Samples were resolved by SDS-PAGE, transferred to nitrocellulose membranes and analyzed for protein AMPylation using 0.13 μg/ml α-AMPylation (clone 1G11) and 0.3 μg/ml α-His antibodies.

### Western blotting

After the samples were resolved by SDS-PAGE and transferred to a nitrocellulose membrane, total proteins were visualized with Ponceau S staining. The membranes were rinsed with TBST buffer (10 mM Tris, pH 8, 150 mM NaCl, 0.1% Tween-20). The membranes were blocked with 5% milk in TBST for 1 h. The membranes were rinsed with TBST. The membranes were incubated with primary antibodies for 1 h at room temperature or overnight at 4 °C. Following incubation with primary antibodies, membranes were washed with TBST and incubated with secondary antibody for 1 h at room temperature. Membranes were washed with TBST and incubated with ECL western blotting substrate for detection using film or imaged with LI-COR Odyssey western blot imaging system.

### Dot blot assay

Samples for dot blot were diluted to 80 ng/μl. Each sample was then serially diluted in a 1:1 manner to prepare samples at 80, 40, 20, 10 and 5 ng/μL. 5 μL of each sample was spotted on nitrocellulose membranes and dried for 1 h. Membranes were equilibrated in TBST and blocked with 5% milk for 30 min. Each membrane was incubated for 1 h with their respective GST-hinT homolog diluted in TBST to 0.125 μg/mL. GST-hinT membranes were washed 3 times for 10 min with 10 mL TBST and incubated with 0.125 μg/mL rabbit α-GST antibody overnight. Membranes were washed 3 times with 10 mL TBST for 10 min and incubated with 0.05 μg/mL IRDye800 α-rabbit antibody for 1 hr at room temperature and washed again 3 times with 10 mL TBST for 10 min. Membranes were then rinsed with TBS and imaged on the LI-COR Odyssey M instrument. For the detection of ADP-ribosylated proteins, membranes were incubated with 0.33 μg/ml α-ADP ribosylation antibodies (Cell Signaling 83732S) overnight at 4 °C, followed by western blotting methods described above.

### Far western analysis

Far westerns for AMPylated and non-AMPylated SodA, Rab1, Rac1 and BiP were performed by resolving 0.5 μg/lane of substrates by SDS-PAGE. Lysate ADP ribosylation, far westerns were performed by diluting ADP ribosylation assay samples described above by 1:8 for no enzyme and PARP1 + HPF, and 1:20 for PARP14 reaction. 5 μL of diluted ADP-ribosylation reactions and 250 ng of AMPylated and non-AMPylated sodA were resolved by SDS-PAGE. Proteins were transferred to a nitrocellulose membrane and blocked with 5% milk for 30 minutes. Membranes were then incubated with 0.06 μg/ml GST-hinT homologs for 1 h at room temperature. Membranes were washed 3 times with 10 mL TBST for 10 min and incubated overnight with α-GST antibody at 4 °C. Alternatively, membranes were incubated with 0.33 μg/ml α-ADP ribosylation or 0.06 μg/ml α-AMPylation (clone 17G6) antibodies overnight at 4 °C. Membranes were washed 3 times

with 10 mL TBST for 10 min and immunoblotted with LI-COR IRDye α-rabbit or α-mouse antibody linked to fluorophore for 1 h at room temperature and washed again 3 times with 10 mL TBST for 10 min. Membranes were then rinsed with TBS and imaged on the LI-COR Odyssey M instrument.

### Cell culture

YUMM3.3 (*Braf^V600E/wt*; *Cdkn2^-/-*) cell lines were obtained from and authenticated by ATCC. Cells were confirmed to be mycoplasma-free using the Mycoplasma detection kit (SouthernBiotech 1310001). YUMM3.3 cells were cultured in DMEM: F12 (Corning MT 10-090CV) supplemented with 10% FBS and 1% penicillin/streptomycin. HEK293a cells were cultured in DMEM (Gibco 11965118) supplemented with 10% FBS and 1% penicillin/streptomycin. Cells were maintained in cell culture incubator with 5% carbon dioxide ($CO_2$) (atmospheric oxygen) and split 2-3 days with fresh media.

### CRISPR editing of *SelO* in mouse melanoma cells

SelO-deficient YUMM3.3 was generated using CRISPR/Cas9[16]. Approximately $1 \times 10^5$ cells were plated adherently in tissue-culture-treated 6-well plates containing DMEM: F12 with 10% FBS and 1% penicillin/streptomycin. 800 ng of each of the sgRNA constructs was co-transfected with 200 ng of eGFP-c1 into the cells using PolyJet (product SL100688, SignaGen) according to the manufacturer's instructions. Growth media was replaced 5 hours post-transfection. After 48 h, one GFP + cell was sorted into each well of a 96-well plate containing DMEM: F12 with 10% FBS and 1% penicillin/streptomycin. The clones were then grown in culture, and SelO protein expression was analyzed by immunoblotting. Genomic DNA from clones was screened to confirm the SelO deletion resulting in YUMM3.3ΔSelO.

### Transfection and immunoprecipitation

**HEK293a cell co-expressing Flag tagged substrates and either SelO, D338A or V334A.** HEK293a cells were plated at $1 \times 10^5$ cells per well in tissue-culture-treated 6-well plates containing DMEM high glucose with 10% FBS and 1% penicillin/streptomycin. 800 ng of flag-tagged substrates cloned into pCCF and 200 ng of SelO U667C or SelO U667C D338A pQXCIP was co-transfected into cells using Polyjet (product SL100688, SignaGen) according to the manufacturer's instructions. Growth media was replaced 5 h post-transfection. After 48 h, cells were placed on ice and washed twice with ice-cold phosphate-buffered saline. 250 µl of lysis buffer (50 mM Tris pH 7.5, 150 mM NaCl, 1% TX100, 1x protease inhibitor cocktail, 1 mM EDTA, 1 mM EGTA) was added to each well. Plates were nutated at 4 °C for 20 min with gentle rocking. Lysate was collected into 1.5 mL tubes and cleared at 21000 x g for 10 min at 4 °C. One tenth of the resulting supernatant was used as cleared cell extract input for immunoblotting. The remaining supernatant was added to 20 µl of pre-washed flag bead suspension. Samples were gently rotated 16–18 h at 4 °C. Beads were spun down 5000 x g for 30 s at 4 °C and washed three times with wash buffer (10 Tris pH 8, 150 mM NaCl, 0.1% TX00, 1 mM EDTA). After final wash, samples were eluted by incubation with elution buffer (10 Tris pH 8, 150 mM NaCl, 0.1% TX00, 1 mM EDTA, 1x flag peptide) for 15 min at room temperature. Elution was repeated twice, and fractions were pooled. SDS loading buffer was added to the pooled elution, which were subsequently resolved by SDS-PAGE and transferred to nitrocellulose membrane for immunoblotting.

**YUMM3.3ΔSelO cells expressing SelO.** For stable expression in melanoma cells, SelO U667C (denoted as SelO) or SelO U667C D338A (catalytically inactive) was cloned into pQCXIP (Clontech). Glud1 was cloned into pMX-IRES-Blasticidin (Cell biolabs) with a C-terminal Flag tag. HEK293 cells were co-transfected with the pQCXIP-SelO U667C or pQCXIP-SelO U667C D338A retroviral vector and the pCL10A1 packaging vector for retrovirus production. The media was changed 5 h after

transfection to reduce cell death from the PolyJet transfection reagent. Viral media supernatant was collected 48 h post-transfection, passed through a 0.45 μM filter, mixed with 8 µg/mL polybrene and added to YUMM3.3ΔSelO. The virus was removed after 2 days, and the infected cells were passaged and selected with 1 µg/mL puromycin.

**YUMM3.3ΔSelO cells co-expressing Glud1-Flag and either SelO or SelO D338A.** HEK293 cells were co-transfected with the pMX-Glud1-Flag, psPAX2, and pMD2G for retrovirus production. YUMM3.3ΔSelO stably expressing SelO were then infected using viral supernatant as described above. The virus was removed after 2 days, and the infected cells were passaged and selected with 5 µg/mL blasticidin.

### GST- hinT H101N Enrichment from cellular lysates

**Preparation of GST-hinT H101N bead resuspension.** For each reaction, 2–4 µg of GST-Ec hinT H101N was nutated with 10 µl bed volume of glutathione beads in 10 mM Tris, pH 8, 150 mM NaCl at 4 °C for approximately 1 h. Beads were centrifuged at $5000 \times g$ for 30 s at 4 °C and washed twice with 10 mM Tris, pH 8, 150 mM NaCl.

**Lysate preparation and binding for E.coli.** BL21 cells were transformed with *E.coli* SelO ppSumo and grown in LB media containing 50 µg/ml of kanamycin at 37 °C. Cells were then induced at an $OD_{600}$ of 0.6 with 0.4 mM IPTG and grown for 16–18 h at 22 °C. Alternatively, BW25113 or knockout strains of GlnA, GlnE, or SelO were grown in LB overnight at 37 °C.

Cells were then harvested by centrifugation. Cells were resuspended and lysed in 10 mM Tris-HCl, pH 8, 150 mM NaCl, 0.1% TX-100, 1 mM PMSF, 1 mM DTT by sonication. Lysates were centrifuged at $21,000 \times g$ for 10 min at 4 °C. Supernatant was transferred and normalized to 1 mg/mL (*E. coli* BL21 strains overexpressing SelO) or 5 mg/mL (*E. coli* BW25113 strains). Approximately 500 µg (*E. coli* BL21 strains overexpressing SelO) or 5 mg (*E. coli* BW25113 strains) of normalized lysates were nutated with the prepared GST-hinT H101N bead resuspension at 4 °C for 2 h. Beads were centrifuged $5000 \times g$ for 30 s at 4 °C and washed three times with 10 mM Tris pH 8, 150 mM NaCl, 0.1% TX-100, 1 mM DTT. Beads were resuspended in SDS loading buffer containing 1% β-mercaptoethanol and boiled. Samples were resolved by SDS-PAGE, transferred to nitrocellulose membranes and analyzed for protein AMPylation using 0.2 µg/ml α-AMPylation clone 1G11 or 0.34 µg/ml α-AMPylation clone 17G6. Alternatively, samples resolved by SDS-PAGE and stained with Invitrogen SilverQuest silver staining kit (Catalog no. LC6070). Untagged *E. coli* SelO migrates at ~ 54 kDa, while *E. coli* SelO ppSumo migrates at ~68 kDa (Fig. 1D), reflecting the added molecular weight of the His-Sumo tag.

**Lysate preparation and binding for YUMM3.3.** YUMM3.3ΔSelO cells stably expressing SelO or SelO D338A were lysed in 25 mM Tris, pH 7.5, 150 mM NaCl, 1% Triton-100, 1 mM EDTA, 1 mM EGTA, 1x PIC. Lysates were centrifuged at $12,000 \times g$ for 10 min at 4 °C. Supernatant was transferred and normalized to approximately 2 mg/mL. Approximately 3 mg of lysate was nutated with 3 µg of GST-hinT H101N bound to glutathione beads. Alternatively, 3 mg of lysate was incubated with 3 µg of α-AMP (clone 17G6) bound to protein A beads[20]. Reactions were nutated at 4 °C overnight. After incubation, beads were centrifuged at $5000 \times g$ for 30 s at 4 °C and washed three times with 10 mM Tris pH 8, 150 mM NaCl, 0.1% TX-100. Beads were resuspended in SDS loading buffer containing 1% β-mercaptoethanol and boiled. Samples were resolved by SDS-PAGE, transferred to nitrocellulose membranes and analyzed for protein AMPylation using 0.65 µg/ml α-AMPylation (clone 1G11).

YUMM3.3ΔSelO cells stably expressing SelO, SelO D338A, or SelO V334A were lysed in 25 mM Tris pH 7.5, 150 mM NaCl, 1% Triton-100, 1 mM EDTA, 1 mM EGTA, 1x PIC, 1 mM DTT. Lysates were centrifuged at $12,000 \times g$ for 10 min at 4 °C. Supernatant was transferred and normalized to approximately 4.5 mg/mL. 9 mg of lysate was nutated with

4 µg of GST-hinT H101N bound to glutathione beads. Reactions were nutated at 4 °C overnight. After incubation, beads were centrifuged 5000 × g for 30 s at 4 °C and washed three times with 10 mM Tris, pH 8, 150 mM NaCl, 0.1% TX-100, 1 mM DTT. Beads were resuspended in SDS loading buffer containing 1% β-mercaptoethanol and boiled. Samples were resolved by SDS-PAGE, transferred to nitrocellulose membranes and analyzed for protein AMPylation using 0.65 µg/ml α-AMPylation (clone 1G11), α-SelO (Abcam EPR11968, 1:1000), α-GAPDH (Thermo-Fisher MA5-15738, 1:20000), α-Glud1 (proteinTech 14299, 1:5000), α-pdhB (proteinTech 14744, 1:3000), α-acat1 (proteinTech 16215, 1:1000), or α-OGDH (proteinTech 66285, 1:3000).

### Sequence alignment
ClustalW multiple sequence alignment of hinT from *E.coli* (CAD6016700.1), *T. marianensis* (WP_013496560.1), *H. sapiens* (NP_005331.1) were performed using MacVector. *E. coli* hinT was cloned with ATG replacing the endogenous TTG to encode the initiator methionine.

### Structural alignment
Structural alignment was done utilizing crystal structures of *E. coli* hinT bound to GMP (3N1S), *H. sapiens* hinT bound to AMP (5KLZ) and *T. marianensis* hinT alpha-fold model bound to AMP. *E. coli* and *T. marianensis* hinT models were aligned using the cealign command on Pymol to the *H. sapiens* model. RMSD obtained from *E. coli* hinT dimer to *H. sapiens* hinT dimer was 2.02 Å over 224 residues. RMSD obtained from *T. marianensis* hinT dimer to *H. sapiens* hinT dimer was 1.15 Å over 224 residues.

### Glud1-Flag immunoprecipitation
YUMM3.3ΔSelO cells co-expressing Glud1-Flag or Glud1-Flag and SelO or SelO D338A were resuspended in lysis buffer containing 50 mM Tris pH 8, 150 mM NaCl, 1% TX-100, 1X Roche cOmplete protease inhibitor, 1 mM EDTA, and nutated at 4 °C for 20 min. Lysates were centrifuged at 12,000 x g for 5 min, and supernatant was extracted and normalized to 1 mg/mL. A sample was collected for the Glud1 activity assay, the rest of the normalized sample was incubated with 10 µL α-Flag M2 affinity agarose overnight at 4 °C. After incubation, beads were centrifuged 5000 × g for 30 s at 4 °C and washed three times with TBSTX-100. Glud1-Flag was eluted by incubating for 10 min with 20 µL of TBSTX-100 containing 1x Flag peptide. Elution was repeated 4 times, and the eluted proteins were pooled. Samples were resolved by SDS-PAGE, transferred to nitrocellulose membranes and analyzed using α-AMPylation (clone 1G11) and α-flag (Sigma F7425, 1:5000). Alternatively, samples were resolved by SDS-PAGE and processed for mass spectrometry analysis of AMPylation sites.

### Glud1 activity assay
Glud1 activity assay was performed utilizing and following the instructions of the Abcam GDH activity assay kit (ab102527). Each reaction contained 100 µL of master reaction mix (GDH assay buffer, GDH developer, and 133 mM glutamate) and 50 µL of 2 mg/mL lysate (prepared similar to Glud1-flag immunoprecipitation) in a 96-well clear plate. Assay was incubated at 37 °C and absorbance at 450 nm was measured at 51 min. All samples were run in technical triplicate. Absorbance at 450 nm of the no lysate control was subtracted from the samples and plotted using GraphPad Prism. The assay was repeated in biological triplicate.

### Metabolomics
YUMM3.3ΔSelO cells stably expressing SelO or SelO D338A were cultured to 50% confluency in DMEM (Fisher Scientific 11965118) supplemented with 10% FBS, 1% penicillin/streptomycin. A 10 cm dish containing adherent cells were gently rinsed with ice-cold saline solution. Dishes were placed on dry ice, and 500 µL of 80%

acetonitrile was added to cells. Cells were incubated on dry ice for 5 min and scraped into an Eppendorf tube. Cells were subjected to three freeze thaw cycles, and then centrifuged at 21,000 x g for 15 min at 4 °C. Supernatant was transferred to a new tube and normalized for protein content. Targeted metabolomics analysis was performed at UT Southwestern CRI Metabolomics facility as previously described[61].

### LC-MS/MS Detection of AMP
AMP levels were monitored by LC-MS/MS using an AB Sciex (Framingham, MA) 6500 + QTRAP mass spectrometer coupled to a Shimadzu (Columbia, MD) Nexera LC. AMP was detected with the mass spectrometer in positive ESI MRM (multiple reaction monitoring) mode by following the precursor to fragment ion transition 348.1 to 139.5. A Thermo Fisher Scientific (Waltham, MA) Biobasic AX anion exchange column (5 micron, 50 × 2.1 mm) was used for chromatography with the following conditions: Buffer A: 8:2 dH$_2$O:MeCN + 10 mM NH$_4$OAc pH 6, Buffer B: 7:3 dH$_2$O:MeCN + 1 mM NH$_4$OAc pH 10.5 with gradient conditions 0–1.0 min 100% A, 1.0–2.5 min gradient to 35% B, 2.5–5.0 min 35% B, 5.0–7.0 min gradient to 65% B, 7.0–8.0 min 65% B, 8.0–8.5 min gradient to 100% B, 8.5–9.5 min 100% B, 9.5–10 min gradient to 100% A, 10 – 11 min 100% A. Uridine-$^{13}$C$_9$, $^{15}$N$_2$-5'monophosphate (transition 336.092 to 102.0) from Sigma (St. Louis, MO) was used as an internal standard (IS). All three GST-tagged hinT homologs were diluted to 20 and 50 µM in 10 mM Tris, 150 mM NaCl, 2 mM DTT. The *E. coli* homolog was also evaluated at 126 µM and the *T. Marianensis* homolog at 98 µM. Protein was denatured to release bound AMP by the addition of a 4x volume of methanol containing 5 µM $^{13}$C$_9$, $^{15}$N$_2$-UMP-IS. Samples were vortexed, incubated for 10 min at RT, and then centrifuged at 16,400 x g for 5 min at 4 °C. The supernatant was analyzed for AMP levels by LC-MS/MS as described above using a standard curve prepared in the buffer used for sample dilution, spiked with known concentrations of AMP (Sigma) and processed the same as samples. Prior to the final analysis, standards made in non-AMP protein-containing buffer versus buffer only were evaluated to ensure the protein matrix did not affect the efficiency of AMP ionization. Samples were injected in duplicate or triplicate at each of the different amounts of hinT protein, sample concentrations of AMP were normalized to the protein concentration, and then the resulting values averaged for each homolog.

### Reporting summary
Further information on research design is available in the Nature Portfolio Reporting Summary linked to this article.

## Data availability
The data supporting the findings of this paper are available within the article and its supplementary information. The source data underlying Figs. 3, 4, 6 and Supplementary Figs. 2, 3, 5, 6, 8–12, and 14 are provided as Source data file. Proteomics data have been deposited in MassIVE MSV000096576. Metabolomics data are available in Supplementary Data 3. The cryo-EM maps have been deposited in the Electron Microscopy Data Bank (EMDB) under accession codes EMD-42892 (Composite map of AMPylated GlnA bound to hinT); and EMD-42896 (GlnA dodecamer with AMPylation). The atomic coordinates have been deposited in the Protein Data Bank (PDB) under accession PDB 8V22 (GlnA dodecamer with AMPylation) and 8V1Y (Composite map of AMPylated GlnA bound to hinT). Previously published PDB structures referred to in this manuscript include 3N1T, 3N1S, 5KLZ, and 7W85. Source data are provided in this paper.

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

## Acknowledgements

A.S. is a W.W. Caruth, Jr. Scholar in Biomedical Research, Cancer Prevention and Research Institute of Texas (CPRIT) Scholar, and a Charles and Jane Pak Center for Mineral Metabolism and Clinical Research Faculty Scholar. This work was supported by NIH Grant K01DK123194 (A.S.), CPRIT Grant RR190106 (A.S.), Welch (I-2046-20200401) and Welch (I-2046-20230405). We thank Drs. Duojia Pan, Zhe Chen, Orson Moe, Vincent Tagliabracci, Chad Brautigam, Kim Orth, Ralph Deberardinis, Lauren Zacharias, Glen Liszczak, and Dan Kober for helpful discussions and reagents. For the HPLC experiments, we acknowledge the effort and resources provided by the institutionally supported UTSW Preclinical Pharmacology Core. For support with Cryo-EM studies, we thank the Structural Biology Lab and the Cryo Electron Microscopy Facility at UT Southwestern Medical Center, which are partially supported by CPRIT grant RP220582. The Children's Research Institute Metabolomics Core is supported by an award from The Cancer Prevention Research Institute of Texas (CPRIT Core Facilities Support Award RP240494).

## Author contributions

A.G. and A.S. designed the experiments. A.G., A.P., Y.H., and A.S. performed the experiments. K.S. performed the mass spectrometry. K.P. performed the bioinformatic analysis. A.G. and A.S. wrote the manuscript with input from all authors.

## Competing interests

The authors declare no competing interests.
