## [Transparent Peer Review file · Nature Communications]

A repurposed AMP binding domain reveals mitochondrial protein AMPylation as a regulator of cellular metabolism

Corresponding Author: Dr Anju Sreelatha

Version 0:

Reviewer comments:

Reviewer #1

(Remarks to the Author)

The manuscript by Abner Gonzalez et al has developed a new method for enriching AMPylated proteins that does not rely on ATP analogues, using the nucleotide-binding protein hinT. This method was validated by demonstrating its ability to directly enrich endogenous AMPylated proteins from *E. coli* lysate. Further structural analysis of hinT and AMPylated GlnA revealed the binding mechanism of hinT to the AMP group. Subsequent homolog analysis identified human and *T. maritima* homologs with higher affinity and selectivity for AMPylation sites. Additionally, compared to AMPylation antibodies, hinT proved to be more efficient at enriching AMPylated proteins from cell lysates, making it applicable to various biological techniques, including pull-down assays and AMPylated protein purification.

Using this technique, the author enriched SelO substrates involved in mitochondrial metabolic pathways in melanoma cell lines overexpressing SelO. Among these substrates was Glud1, where the antenna domain Y464 could be AMPylated by SelO, resulting in decreased catalytic activity. This suggests that SelO-mediated AMPylation can regulate Glud1's function, though its physiological significance remains to be further clarified.

AMPylation plays an important role in the homeostasis of both eukaryotic and prokaryotic cells, as well as in bacterial infection processes. This study provides a new method for identification of the substrates of AMPylases, which currently is still a challenge in AMPylation research. This paper should be considered for Nature Communications after major revisions. Several biochemical data lack the controls. I have the following questions.

1. In Fig. 3, do hinT homologs has low binding affinity to unmodified sucA? Can the differences in affinity between hinT homologs for AMPylated-protein be observed for other AMPylated proteins, not just AMP-SucA?
2. Does replacing E96 with T or S enhance *Ec* hinT's affinity for AMPylated proteins? Additionally, how do the variable C termini of the hinT homologues in Fig. 3 affect their affinities for AMPylate proteins?
3. The authors mentioned that human hinT has strong binding affinity to AMP and is not suitable for binding and pulling down the AMPylated substrates from cell lysates. However, in Figure 4a, human hinT protein has stronger binding affinity to AMPylated proteins, suggesting human hinT protein likely is better than other homologues as a tool.
4. Fig. 4a should contain a control of the unmodified GlnA protein.
5. Can recombinant SelO directly modify the top substrate candidates shown in Fig. 5c, not only Glud1, in vitro?
6. Fig 6D and 6F lack the controls.
7. In Fig 6D, why does Glud1 have two bands on SDS-PAGE?
8. In Fig 6F, the activity of AMPylated Glud1 should be tested by using recombinant proteins after AMPylation by SelO in vitro.
9. Glud1 AMPylation was only detected in SelO-overexpressing YUMM3.3 cells. Can the AMPylation of Glud1 be detected under any physiological conditions?
10. Does Glud1 has AMPylation sites on other residues, not just tyrosine, by SelO? The authors used *Ec* hinT in the pull down assays, which is possibly the reason why they only found the tyrosine AMPylation in Glud1 by SelO.

Reviewer #2

(Remarks to the Author)

This study by Gonzalez et al. introduces a first-in-kind capturing strategy to selectively enrich AMPylated proteins from complex cell lysates. Guided by structural considerations, the authors mutagenize the nucleotide binding protein, hinT, for it

to better recognize and bind AMPylated proteins. The authors use this new tool to identify novel targets for the mitochondrial AMPylase, SelO and demonstrate that SelO-mediated AMPylation of mitochondrial proteins regulates metabolic flux by AMPylating enzymes such as GLUD1.

My enthusiasm for this manuscript is high. The paper is well written in clear language, does not over-interpret results, and has a logic structure that makes it easy to read and follow. The figures are well designed and support both the narrative and the presented conclusions. The presented approach to catch AMPylated proteins using mutant hinT will be of great interest to the field and might inspire related efforts to design proteins binding other PTMs such as UMPylated residues.

As you will see below, I have only one major point of concern, which I hope and assume will be straightforward for the author to address, and a couple of minor suggestions.

Major point:

Reproducibility of results. This concerns Figures 1B-E, 4A-D, 5B, 6C-D. The blots presented in all these figures are of high quality and support the author's conclusion without question. However, all of these blots represent an n of 1. There is no indication anywhere in the paper whether these blots were replicated at least three times to confirm the presented results. In some cases, such as for figure 1B-E, one could argue that these findings were replicated again in other experiments presented in this study, and thus do not need to be repeated as presented. For the results presented in figure 4, however, it is critical to show that these dose-dependent binding effects replicate. The same applies for Figure 6D. I thus ask the authors to consider the following changes: 1) disclose in the figure legends how often each experiment/blot was replicated and indicate where the additional replicates can be found; 2) disclose all additional replicates of these blots as supplementary data; 3) for figure 4A-C, quantify signal intensities in all three replicates and disclose the results in a bar/line graph in the main figure or supplement. Assuming that the authors already performed each experiments multiple times to confirm their results, this should be a minor effort.

Minor points:

Figure 1D: Please label SelO OE band, too (I assume it's the one running at 70kDa?)

Figure 1D: Do you have any idea what the prominent band around 55kDa in the D256A/GST-hinT lane could be? It's also picked up in the SelO wt, but not the V242A OE samples. And, based on Fig. 1E, it's probably AMPylated (maybe GlnA? But why wouldn't it be modified in the V242A sample?).

Figure 2: please add the obtained structural resolution to the figure legend.

Figure 2: Could the authors comment on whether or not hinT binding to GlnA induces a conformational change in the dodecamer?

Figure 5C: increase font size of all labels (axis and points)

Figure S8A: The presented results indicate that none of the hinT versions binds to BiP-AMP. Could the authors speculate why this might be the case?

Figure S8B-C: The authors claim that Tm hinT doesn't pick up ADP-rybosylation. However, there is a faint yet clearly visible signal on the Tm hinT-probed dot blot membrane in lanes 1-3, which represent ADP-ribosylated samples. Further, in Figure S8C, there is a clear band in the lysate & PARP14 sample. The authors might consider adding a statement that Tm hinT also shows residual binding activity towards high-abundant ADP-ribosylated proteins.

Signed Matthias Truttmann & Corey Stewart.

Corey Stewart: I co-reviewed this manuscript with one of the reviewers who provided the listed reports. This is part of the Nature Communications initiative to facilitate training in peer review and to provide appropriate recognition for Early Career Researchers who co-review manuscripts.

Reviewer #3

(Remarks to the Author)

Reviewer #4

(Remarks to the Author)

AMPylation is a post-translational modification where an adenosine monophosphate (AMP) group is covalently attached to a protein, often altering its activity, localization, or interactions. It plays a crucial role in cellular processes such as signal transduction, bacterial virulence, and the regulation of protein function, making it significant for understanding diseases and developing targeted therapeutics.

In the present manuscript, the authors present a capture method to identify new targets of AMPylation, based on histidine triad nucleotide binding protein (hinT) serving as AMP binding domain. By cryoEM, they clarify AMPylated protein bound to the hinT nucleotide-binding pocket, revealing substrate recognition and specificity. The structure is used to optimize the approach. Based on that they authors screen in mouse melanoma cell lines, and identify several new mitochondrial proteins

as putative targets of AMPylation. It appears that the role of this modification is within the regulation of metabolic flux.

Overall, this is a well-written and concise manuscript with carefully executed experiments. It provides a solid basis for future research into the functional roles and mechanisms of protein AMPylation. We recommend publication after fixing some minor issues.

Minor:

- The choice of hinT for identifying AMPylated proteins is introduced in a comprehensible way; however, in my opinion, some questions remain unanswered. For instance, it would be interesting to discuss how hinT compares to other nucleotide-binding proteins—what qualifies hinT as the optimal candidate for this method? In this context, it would also be valuable to address whether the method can be applied to other nucleotide-binding ('bait') proteins. Maybe, the authors can add one or two sentences to address that?

- It would be helpful to provide more detail on the non-specific bindings of hinT to better validate the specificity of the method. A more in-depth discussion of the repeatedly observed band at 70 kDa (e.g., Figures 1C-E) would also be interesting.

Fig.5: The stars marking the bands at 25 kDa and 55 kDa in Figure 5 are not explained in the figure legend. There is only a note about them in the main text. For better understanding, it would be helpful to include a brief explanation of the markings in the figure legend.

Reviewer #5

(Remarks to the Author)

Version 1:

Reviewer comments:

Reviewer #1

(Remarks to the Author)

The authors have addressed all my concerns. It should be accepted for publication.

Reviewer #2

(Remarks to the Author)

The authors have addressed all our concerns. At this point, I'd like to congratulate the authors to their nice work and support acceptance of the revised manuscript for publication in Nature Communications.

Signed Matthias C Truttmann and Corey Stewart

Reviewer #3

(Remarks to the Author)

Reviewer #4

(Remarks to the Author)

The authors have addressed all questions and concerns raised. We would like to congratulate them on this excellent piece of work.

Reviewer #5

(Remarks to the Author)

Manuscript title: **A repurposed AMP binding domain reveals mitochondrial protein AMPylation as a regulator of cellular metabolism**

We thank the reviewers and the editor for providing valuable suggestions that have strengthened this manuscript. Below, you will find a point-by-point reply to each of the reviewers' comments, with our responses presented in blue.

Figures that have been added to the revised manuscript are denoted as "Revised figure X". Figures that are shown only in the author responses, but not in the manuscript, are denoted as figures R1- R6.

Reviewer 1

1. In Fig. 3, do hinT homologs has low binding affinity to unmodified sucA? Can the differences in affinity between hinT homologs for AMPylated-protein be observed for other AMPylated proteins, not just AMP-SucA?

We used biolayer interferometry to analyze the binding affinity of unmodified sucA to hinT homologs (**Revised Supplemental figure S5D**). *E. coli*, *T. marianensis*, and human homologs of hinT^{HN} displayed binding to AMPylated sucA similar to those observed in Figure 3. However, hinT homologs did not bind to unmodified sucA. These results are in line with Figure 4D which demonstrates unmodified sucA does not bind to hinT.

Figure S5D: Representative BLI sensorgrams depicting the binding response of 500 nM AMPylated sucA or unmodified sucA to immobilized GST-*E. coli* hinT, GST-*T. marianensis* hinT, and GST-*H. sapiens* hinT.

To analyze the affinity of hinT homologs to other AMPylated proteins, we performed BLI using AMPylated GlnA and *E. coli* and *T. marianensis* hinT. We observed a significant difference in binding affinity, with *T. marianensis* hinT binding to GlnA-AMP with higher affinity than *E. coli* hinT (**Figure R1**). We could not determine a binding affinity because GlnA forms a stable dodecamer, which allows for various permutations of up to 12

equivalents of AMP per molecule. Furthermore, the binding did not saturate even at a concentration of 2500 nM GlnA.

Figure R1: Steady state binding response of GlnA-AMP to immobilized GST-hinT homologs from BLI experiment

Although not as quantitative as BLI, the data in Figure 4A and 4B demonstrate the differences in affinity between hinT homologs for several AMPylated proteins. For example, Rac1-AMP displays strong binding to the human homolog of hinT, whereas Rab1-AMP shows the strongest binding to the *T. marianensis* homolog. The binding of each AMPylated substrate will need to be quantitatively assessed in future studies.

2. Does replacing E96 with T or S enhance *Ec* hinT's affinity for AMPylated proteins? Additionally, how do the variable C termini of the hinT homologues in Fig. 3 affect their affinities for AMPylate proteins?

We performed BLI to demonstrate that *E. coli* hinT^{HN} binds to sucA-AMP with a K_D of ~30 nM, while E96S mutation binds with K_D ~ 21 nM (**Revised Supplemental figure 8**). This modest improvement in binding may contribute the differences in binding observed in Figure 4A and Figure 4B with AMPylated substrates such as GlnA or Rac1.

Figure S8

Figure S8: Mutation of E96S improves binding affinity of *E. coli* hinT

(A-B) Representative BLI sensorgrams depicting the binding response of serial dilutions 2500 nM to 3.4 nM AMPylated sucA to immobilized GST-*E. coli* hinT (A), GST *E. coli* -hinT^{E96S} (B).

(C) Binding affinities measured from steady state binding response of GST-*E. coli* hinT and GST *E. coli* -hinT^{E96S} to sucA-AMP.

The C terminus is hypothesized to play an important role in the specificity and affinity of hinT proteins^{1,2}. The *E. coli*, human, and *T. marianensis* homologs share about 50% sequence identity with significant variations in the C terminus. Specifically, the C-terminal sequence of *H. sapiens* hinT is more similar to that of *T. marianensis* than to *E. coli*. We observed that *T. marianensis* and *H. sapiens* bind more tightly to sucA (Figure 3F), yet they exhibit different substrate specificities (Figure 4A & 4B).

While we initially used the C terminal divergence to guide our investigation of hinT, other amino acids also influence binding affinities. To address the reviewer's question regarding the functional role of the C terminus, we attempted to create chimeric proteins by swapping the C-terminal amino acids. However, the resulting proteins exhibited low yield and poor stability, rendering them unsuitable for further assays. Future studies will optimize the construction of C terminal variants to better understand their role in binding affinities.

3. The authors mentioned that human hinT has strong binding affinity to AMP and is not suitable for binding and pulling down the AMPylated substrates from cell lysates. However, in Figure 4a, human hinT protein has stronger binding affinity to AMPylated proteins, suggesting human hinT protein likely is better than other homologues as a tool.

We agree with the reviewer's comment that human hinT is likely a better tool for the detection of AMPylated proteins. Human hinT has strong affinity for AMPylated substrates as seen in Figures 3F, 4A and 4B. However, human hinT also shows the highest affinity for AMP among all homologs (Supplemental Fig S5E). High concentration of AMP nucleotide can compete with AMPylated substrates as seen in Figure 4C. Thus, we reason that human hinT is a valuable tool for enriching AMPylated substrates in cell-free systems and for detection (far-western blots) but may be limited in enrichment from cellular lysates that may contain more AMP than AMPylated proteins.

4. Fig. 4a should contain a control of the unmodified GlnA protein.

Glutamine synthetase (GlnA) is regulated by multiple mechanisms to balance its activity in response to cellular status. One such mechanism is the enzyme's dodecameric structure, which allows for up to 12 equivalents of AMP per molecule. During purification of his tagged GlnA from *E. coli* BL21, we observe a mix of AMPylated and non-AMPylated species. We attempted to separate unmodified GlnA from GlnA-AMP using stringent purification methods as well as altering the growth conditions in *E. coli*. Although we were unsuccessful at generating the unmodified GlnA, we acknowledge that the effective concentration of AMPylated GlnA is lower than initially noted in our figures and methods due to the formation of the dodecamer with AMPylated and non-AMPylated species. Mass spectrometry revealed that our GlnA-AMP contains ~48% AMPylated species.

5. Can recombinant SelO directly modify the top substrate candidates shown in Fig. 5c, not only Glud1, in vitro?

We performed AMPylation assays with recombinant SelO, Glud, and biotin ATP. In vitro, purified SelO displays robust auto-AMPylation but low AMPylation of human (Hs) and bovine (Bt) Glud (**Figure R2**). Note the ratio of AMPylated Glud and SelO (top panel: α -avidin) in comparison the total amount of protein loaded (bottom panel: ponceau stained membrane). Mass spectrometry analysis revealed an AMPylation stoichiometry of only 4% in vitro, consistent with our observations that recombinant human SelO exhibits low AMPylation activity in vitro. The reduced activity may result from missing cofactors or suboptimal conditions for enzyme activity.

Figure R2: Representative blot using avidin HRP to detect biotinylated proteins following incubation of human (Hs) or bovine (Bt) Glud with bio-17-ATP and human SelO or the D338A mutant.

To address the reviewer’s question regarding SelO’s ability to modify the substrates shown in the volcano plot, we co-expressed SelO, inactive D338A, or hyperactive V334A SelO with flag tagged substrates in HEK293a cells. We performed flag immunoprecipitation and assessed AMPylation of each substrate using monoclonal α -AMP antibodies. SelO and the hyperactive V334A mutant, but not the inactive D338A, AMPylates multiple mitochondrial proteins (**Revised Figure S14**).

Figure S14: Protein immunoblotting of flag immunoprecipitates or cell lysates from HEK293a cells co-expressing SelO and flag tagged substrates: mitochondrial aconitase (aco2), pyruvate dehydrogenase kinase (pdk1), short-chain specific acyl-CoA dehydrogenase (acadS), pyruvate dehydrogenase E1 component subunit α (pdhA1), succinate dehydrogenase flavoprotein subunit (sdhA), pyruvate dehydrogenase E1 component subunit β (pdhB), isobutyryl-CoA dehydrogenase (acad8). Control (lane 1) denotes non-transfected HEK293a cells.

Note that a non-specific band is observed with SdhA co-expressed with inactive SelO. Mass spectrometry analysis has confirmed that SdhA is not AMPylated when co-expressed with inactive SelO.

Transient transfection of SelO results in two SelO immunoreactive bands. We have performed subcellular fractionation to show that the mitochondrial SelO corresponds to the lower migrating species and the cytoplasmic SelO corresponds to the higher migrating due to the presence of mitochondrial targeting sequence. Overexpression of mitochondrial proteins can lead to some cytosolic mis-localization due to import machinery saturation (Schmidt et al. PMID 20729931).

6. Fig 6D and 6F lack the controls.

We have performed the assays in triplicate with controls and included the results in **Revised Figure 6D and 6F, supplemental figure S14B and S14C.**

Figure 6D. Protein immunoblotting of flag immunoprecipitates or cell lysates from YUMM3.3 Δ SelO expressing Glud1-Flag or Glud1-flag and SelO or SelO^{D338A}. Results are representative of at least 3 independent experiments (Supplementary Fig. S14b and S14c).

The methods section has been revised to indicate that the glutamate dehydrogenase activity assay was performed using a kit from a new vendor (Abcam) as the previous vendor (Sigma) discontinued the item. The assays were performed in technical and biological triplicates.

Figure 6F. The activity of Glud1 in YUMM3.3ΔSelO expressing Glud1-Flag or Glud1-Flag and SelO or SelO^{D338A}. Data represent the average of 3 technical replicates p<0.001.

7. In Fig 6D, why does Glud1 have two bands on SDS-PAGE?

We have observed that AMPylation can result in mobility shift of some substrates. One example is shown below with *E. coli* superoxide dismutase. Intact mass and LC-MS/MS analysis revealed that sodA is AMPylated on multiple amino acids by *E. coli* SelO in bacteria (Figure R3).

Figure R3: SDS-PAGE and Coomassie blue staining analysis of recombinant *E. coli* sodA.

We hypothesize that the slower migrating species of Glud1 is more AMPylated. Immunoblotting the flag immunoprecipitates reveals that the top band is more reactive to α-AMP antibodies (Figure R4).

Figure R4. Protein immunoblotting of flag immunoprecipitates from YUMM3.3 Δ SeIO expressing Glud1-flag or Glud1-flag and SeIO or SeIO^{D338A}. The blot was probed with mouse α -AMP and rabbit α -flag and then detected with IRDye 800 and IRDye 680 secondary antibodies, respectively.

8. In Fig 6F, the activity of AMPylated Glud1 should be tested by using recombinant proteins after AMPylation by SeIO in vitro.

Please see response to Comment no. 5. We speculate that no changes in activity will be observed due to low stoichiometry of AMPylation in vitro.

9. Glud1 AMPylation was only detected in SeIO-overexpressing YUMM3.3 cells. Can the AMPylation of Glud1 be detected under any physiological conditions?

In this manuscript, we used SeIO-overexpressing cells to demonstrate that hinT can serve as a tool to enrich for AMPylated proteins. We are currently investigating several of the candidate substrates to determine the importance of AMPylation in mitochondrial biology and physiology. Due to the difficulty and variability of endogenous selenoprotein expression in cell culture, we generated SeIO knockout mice to analyze AMPylation under physiological conditions across various tissues. Notably, we identified AMPylation of Glud1 in hepatic tissue of wild-type C57BL/6 mice but not in SeIO knockout mice. We are performing metabolic tracing studies to determine the functional consequences of Glud1 AMPylation in murine liver. However, we believe these studies are outside the scope of this manuscript, they highlight new opportunities and further support the use of hinT as a tool to study AMPylation.

10. Does Glud1 has AMPylation sites on other residues, not just tyrosine, by SeIO? The authors used Ec hint in the pull down assays, which is possibly the reason why they only found the tyrosine AMPylation in Glud1 by SeIO.

We apologize for the miscommunication. Mass spectrometry analysis to identify the site of AMPylation was performed using flag immunoprecipitates, not hinT pulldown assays. We identified three tyrosine residues as potential sites of AMPylation in Glud1 co-expressed with SeIO but not the inactive mutant, D338A. We have revised the figure legend and methods to clarify this point. Furthermore, we agree with this keen observation that the identification of substrates may be limited by the hinT homolog used for enrichment.

Reviewer 2 and 3

Major point: Reproducibility of results. This concerns Figures 1B-E, 4A-D, 5B, 6C-D. The blots presented in all these figures are of high quality and support the author's conclusion without question. However, all of these blots represent an n of 1. There is no indication anywhere in the paper whether these blots were replicated at least three times to confirm the presented results. In some cases, such as for figure 1B-E, one could argue that these findings were replicated again in other experiments presented in this study, and thus do not need to be repeated as presented. For the results presented in figure 4, however, it is critical to show that these dose-dependent binding effects replicate. The same applies for Figure 6D. I thus ask the authors to consider the following changes: 1) disclose in the figure legends how often each experiment/blot was replicated and indicate where the additional replicates can be found; 2) disclose all additional replicates of these blots as supplementary data; 3) for figure 4A-C, quantify signal intensities in all three replicates and disclose the results in a bar/line graph in the main figure or supplement. Assuming that the authors already performed each experiments multiple times to confirm their results, this should be a minor effort.

We thank the reviewers for highlighting this important detail, which will enhance the clarity and robustness of our manuscript. We have performed the assays depicted in the figures of this manuscript in at least triplicate. As requested, we have added the reproducibility of each experiment and provided details on where the additional replicates can be found in the figure legends for Figures 4 and 6D. The following supplemental figures are linked to the replicates.

Fig. 4a and Supplementary Fig. 9
Fig. 4b, Supplementary Fig. 10b-e
Fig. 4c, Supplementary Fig. 11
Fig. 6d and Supplementary Fig. 14b-c

We appreciate the reviewers' suggestion to include the analysis of signal intensities. We have performed the requested analysis for Figures 4A-C and attached the results as a supplemental file in the revision. However, we request to only include one representative analysis in the manuscript for clarity and conciseness.

We would like to note that the raw data from these replicates are highly similar. Additionally, quantifying signal intensities across separate membranes presented challenges due to inherent variations in gels and/or high intensity speckles that would result in misleading signal intensities. We believe that including a representative analysis best reflects the consistency of our findings while maintaining the manuscript focus.

Minor points:

Figure 1D: Please label SelO OE band, too (I assume it's the one running at 70kDa?)

We labeled the band in the figure and added clarification to the figure legend and methods.

Figure 1D: Do you have any idea what the prominent band around 55kDa in the D256A/GST-hinT lane could be? It's also picked up in the SelO wt, but not the V242A OE samples. And, based on Fig. 1E, it's probably AMPylated (maybe GlnA? But why wouldn't it be modified in the V242A sample?).

Mass spectrometry analysis (Supplementary table 2 and Fig.S1b and S1c) suggests that GlnA is the 55 kDa observed in Fig. 1d and 1e. Lower abundance of GlnA is found in hinT enrichment from WT and V242A in comparison to inactive D256A lysates (Figure R5).

Figure R5. Smear plots of full set of proteins identified by quantitative MS/MS analysis edited from Supplementary Fig 1b and 1c to depict de-enrichment of GlnA in lysates over-expressing SelO.

SelO-mediated AMPylated proteins may compete with endogenous GlnA-AMP. As a result, hinT pulldown from lysates expressing WT SelO shows some GlnA enrichment, whereas lysates expressing the hyperactive mutant display little to no pulldown of GlnA, likely due to the high AMPylation present in V242A lysates.

Figure 2: please add the obtained structural resolution to the figure legend.
Structural resolution has been added to the figure legend.

Figure 2: Could the authors comment on whether or not hinT binding to GlnA induces a conformational change in the dodecamer?

Structural alignment reveals that our model closely resembles the previously solved structure of GlnA (PDB:7W85). A notable difference lies in the AMPylation site, which is positioned within a long flexible loop. In our structure, hinT appears to “pull” this loop for recognition and binding (**Figure R6**).

Figure R6. Zoomed in view of the GlnA Tyr398-AMP within the hinT binding pocket in comparison to unmodified GlnA. Tyr398 is colored in yellow.

Figure 5C: increase font size of all labels (axis and points)
We increased the font size for all labels in Fig. 5c

Figure S8A: The presented results indicate that none of the hinT versions binds to BiP-AMP. Could the authors speculate why this might be the case?

We observed that hinT homologs exhibit high affinity for AMPylated tyrosine residues but lower affinity for threonine AMPylated sites, as observed with Rac1-AMP. Since BiP is AMPylated on a threonine residue, this may contribute to its reduced affinity for hinT. Additionally, structural features unique to BiP could further hinder hinT binding. Interestingly, BiP was identified in our mass spectrometry analysis of YUMM3.3 cells. However, BiP was excluded in further analysis because there was no enrichment in SelO vs. SelO DA. Our current studies are aimed at detecting and optimizing enrichment of AMPylation that is independent of SelO.

Figure S8B-C: The authors claim that Tm hinT doesn't pick up ADP-ribosylation. However, there is a faint yet clearly visible signal on the Tm hinT-probed dot blot membrane in lanes 1-3, which represent ADP-ribosylated samples. Further, in Figure S8C, there is a clear band in the lysate & PARP14 sample. The authors might consider adding a statement that Tm hinT also shows residual binding activity towards high-abundant ADP-ribosylated proteins.

We added the sentence to the results section describing Supplementary Fig. 12b: hinT shows little to no reactivity with ADP-ribosylated proteins at the tested concentrations but may exhibit residual binding toward highly abundant ADP-ribosylated proteins.

Reviewer 4 and 5

Minor:

-The choice of hinT for identifying AMPylated proteins is introduced in a comprehensible way; however, in my opinion, some questions remain unanswered. For instance, it would be interesting to discuss how hinT compares to other nucleotide-binding proteins—what qualifies hinT as the optimal candidate for this method? In this context, it would also be valuable to address whether the method can be applied to other nucleotide-binding ('bait') proteins. Maybe, the authors can add one or two sentences to address that?

We appreciate the opportunity to elaborate on this key point. The histidine triad (HIT) superfamily shares some similarities but also some notable differences when compared to other nucleotide binding proteins. To illustrate the differences, we compared the nucleotide binding pockets in the following domains: protein kinase, ATPase, and cystathionine beta synthase. Notably, many nucleotide-binding sites are more deeply buried or less exposed than the binding pocket in hinT. Furthermore, hinT binds the mononucleotide AMP in the proper orientation in a binding pocket that is more open for AMPylated protein binding.

We have added the following text to discussion: Comparison of the binding pocket of hinT with other nucleotide-binding proteins, such as the AMP-binding Cystathionine Beta Synthase (CBS) domain, reveals that hinT binds AMP in the proper orientation and spatial arrangement. While hinT exhibits specific AMP binding, its more accessible binding site provides ample space for the AMPylated protein. This characteristic makes hinT a more promising candidate for specifically binding AMP moieties in AMPylated proteins (**Revised Supplementary Fig. 16**).

Figure S16

Figure S16: Comparison of binding pockets in nucleotide-binding proteins

- It would be helpful to provide more detail on the non-specific bindings of hinT to better validate the specificity of the method. A more in-depth discussion of the repeatedly observed band at 70 kDa (e.g., Figures 1C-E) would also be interesting.

The non-specific 70 kDa band in Fig. 1c results from non-specific binding of the AMPylation antibodies to an unidentified protein in *E. coli* lysates. In Fig. 1d and 1e, the 70 kDa band represents over-expressed SelO. We have added the SelO label to the figure and further clarified this in the figure legends and methods. The non-specific

binding observed in Fig. 1d, lanes 5&6, with inactive D256A SeO is likely due to the interaction of the highly abundant SeO with GST, as this band is also present in GST only control.

hinT demonstrates low nanomolar affinity towards AMPylated proteins. We observed that hinT is highly specific for AMPylated proteins in dot blot and far western analysis. However, the reviewer raises an important question regarding the non-specific binding, especially in the context of enrichment. Our mass spectrometry analysis reveals that hinT can bind other nucleotide binding proteins, such as ATP and RNA-binding proteins, though this can be filtered out using an AMPylation deficient control (Supplementary table S4). We have added the following in the discussion: Additionally, hinT may interact with nucleotide-binding proteins, including RNA binding proteins, but this can be effectively filtered out using an AMPylation deficient control.

Fig.5: The stars marking the bands at 25 kDa and 55 kDa in Figure 5 are not explained in the figure legend. There is only a note about them in the main text. For better understanding, it would be helpful to include a brief explanation of the markings in the figure legend.

We have included the following sentence in the figure legend: * denotes the positions of heavy and light chains of AMP antibodies used for immunoprecipitation.

References

- 1 Chou, T. F., Sham, Y. Y. & Wagner, C. R. Impact of the C-terminal loop of histidine triad nucleotide binding protein1 (Hint1) on substrate specificity. *Biochemistry* **46**, 13074-13079 (2007). <https://doi.org/10.1021/bi701244h>
- 2 Bardaweel, S., Pace, J., Chou, T. F., Cody, V. & Wagner, C. R. Probing the impact of the echinT C-terminal domain on structure and catalysis. *J Mol Biol* **404**, 627-638 (2010). <https://doi.org/10.1016/j.jmb.2010.09.066>

Manuscript title: **A repurposed AMP binding domain reveals mitochondrial protein AMPylation as a regulator of cellular metabolism**

We thank the reviewers for their thoughtful comments and constructive feedback, which have improved the quality and clarity of our manuscript.

REVIEWERS' COMMENTS

Reviewer #1 (Remarks to the Author):

The authors have addressed all my concerns. It should be accepted for publication.

Reviewer #2 (Remarks to the Author):

The authors have addressed all our concerns. At this point, I'd like to congratulate the authors to their nice work and support acceptance of the revised manuscript for publication in Nature Communications.

Signed Matthias C Truttmann and Corey Stewart

Reviewer #3 (Remarks to the Author):

Reviewer #4 (Remarks to the Author):

The authors have addressed all questions and concerns raised. We would like to congratulate them on this excellent piece of work.

Reviewer #5 (Remarks to the Author):
